# Stratify or Die: Rethinking Data Splits in Image Segmentation

**Naga Venkata Sai Jitin Jami**[1,2]   **Thomas Altstidl**[1]   **Jonas Leo Mueller**[1]   **Jindong Li**[1]
**Dario Zanca**[1]   **Björn Eskofier**[1]   **Heike Leutheuser**[2]

[1] Machine Learning and Data Analytics Lab, FAU Erlangen-Nürnberg, Germany
[2] Ambient Assisted Living and Medical Assistance Systems, Department of Computer Science, University of Bayreuth, Germany
{jitin.jami,thomas.r.altstidl,jonas.leo.mueller,jindong.li}@fau.de
{dario.zanca,bjoern.eskofier}@fau.de   heike.leutheuser@uni-bayreuth.de

## Abstract

Random splitting of datasets in image segmentation often leads to unrepresentative test sets, resulting in biased evaluations and poor model generalization. While stratified sampling has proven effective for addressing label distribution imbalance in classification tasks, extending these ideas to segmentation remains challenging due to the multi-label structure and class imbalance typically present in such data. Building on existing stratification concepts, we introduce Iterative Pixel Stratification (IPS), a straightforward, label-aware sampling method tailored for segmentation tasks. Additionally, we present Wasserstein-Driven Evolutionary Stratification (WDES), a novel genetic algorithm designed to minimize the Wasserstein distance, thereby optimizing the similarity of label distributions across dataset splits. We prove that WDES is globally optimal given enough generations. Using newly proposed statistical heterogeneity metrics, we evaluate both methods against random sampling and find that WDES consistently produces more representative splits. Applying WDES across diverse segmentation tasks, including street scenes, medical imaging, and satellite imagery, leads to lower performance variance and improved model evaluation. Our results also highlight the particular value of WDES in handling small, imbalanced, and low-diversity datasets, where conventional splitting strategies are most prone to bias.

## 1   Introduction

Image segmentation involves the assignment of a label to each pixel in an image, with applications in diverse fields such as autonomous driving [1–5], medical imaging [6–10], and satellite image analysis [11–15]. In a supervised learning setting, splitting the dataset into disjoint subsets for training and validation is common practice. However, following a random splitting strategy can result in class distribution shifts across subsets compared to the original dataset. This can lead to biased evaluations, as the test set is typically smaller than the training set and more affected by this phenomenon. Typically, such issues are addressed through stratification to guarantee a consistent distribution of class labels across subsets [16] but this is not straightforward for image segmentation datasets [17–21]. The fundamental unit, an image, is not a sample with a single class but a sample with pixels belonging to many classes. Stratification strategies for similarly complex multi-label datasets [22] could provide interesting solutions that can be applied to segmentation datasets. Strategies relevant to multi-label datasets have been proposed and investigated in various forms ranging from iterative strategies [23–25], greedy search [26, 27], and genetic algorithms [28].These methods do not directly apply to the problem of segmentation, as multi-label datasets have the complexity of multiple positive

39th Conference on Neural Information Processing Systems (NeurIPS 2025).

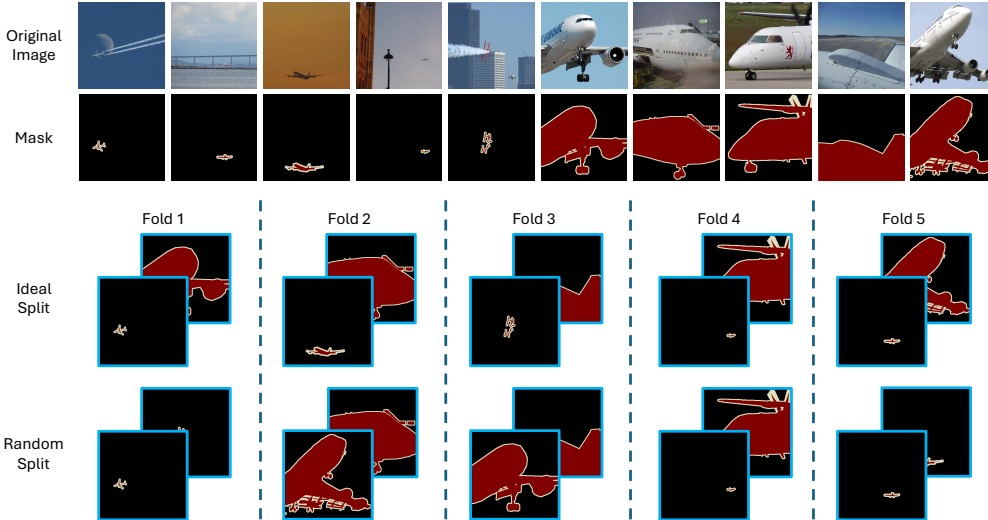

Figure 1: Representative examples of *Plane* images from PascalVOC [17]. The illustration shows an ideal split with balanced class distribution across folds which is not ensured by random splitting. Such balance is essential to avoid bias and ensure reliable model evaluation.

classes for each sample but do not have the added complexity of proportional representation of these classes in the sample. Segmentation datasets present a unique challenge in this regard, complicating the task of stratification.

Segmentation datasets are often small due to high annotation costs and exhibit imbalanced class distributions due to label diversity. As dataset size decreases, inconsistent class distributions in subsets increase, hindering generalization [29]. This issue is critical in imbalanced datasets, where some classes may be absent from splits, making performance metrics like F1-score and IoU unmeasurable. Such missing data undermines the reliability of model evaluation, especially in K-fold cross-validation [30], where unrepresentative subsets affect performance variance.

Motivated by these challenges, we propose and investigate two stratification strategies: *Iterative Pixel Stratification (IPS)* and *Wasserstein-Driven Evolutionary Stratification (WDES)*[1]. These methods are designed to assign samples to subsets, or folds, for a train-test split scenario (two folds) or a K-fold cross-validation scenario ($K$ folds), taking into account the label composition of the sample. IPS extends the Iterative Stratification (IS) algorithm proposed in literature [23], assigning samples to folds using a greedy approach to meet predefined class presence requirements measured by the number of pixels in each fold. It prioritizes assigning samples containing rare classes to folds, progressively addressing those with increasing ubiquity. In contrast, WDES employs an evolutionary algorithm akin to Evosplit [28], where a population of potential fold assignments (individuals) is evaluated simultaneously. The fittest individuals undergo selective crossover and mutations for further refinement. The fundamental distinction between these two methods lies in their approach to fold assignment and evaluation. The iterative nature of IPS confines it to a sequential decision-making process and is limited to a single chain of iterations. Conversely, the genetic algorithm in WDES evaluates multiple solutions independently by maintaining a population, enabling it to explore a broader solution space. The flexibility of WDES allows us to choose a better objective function to be minimized. We choose the Wasserstein distance between the class distribution in the fold and the overall dataset as it indicates the similarity between these distributions. Hence, minimizing the distance reduces dissimilarity between the subsets and the dataset.

We evaluate IPS, WDES, and random splitting across five benchmark segmentation datasets. Our evaluation begins by analyzing the consistency of folds generated through repeated experiments. We

---

[1]Implementation available at: `https://github.com/jitinjami/SemanticStratification`

further assess the impact of targeted stratification versus random splitting by examining the standard deviation of accuracy, F1-score, and Intersection over Union (IoU) across 10-fold cross-validation. A lower deviation indicates a more reliable assessment of model performance. Empirical results show that WDES consistently achieves the highest quality splits. In contrast, IPS under-performs relative to WDES, suggesting that a sequential strategy may be suboptimal for segmentation tasks. Notably, WDES also demonstrates better consistency over the other two for evaluating model performance on low-entropy datasets, where class distributions are highly imbalanced or concentrated. Overall, our findings highlight the importance of stratification tailored to structured outputs and advocate for the adoption of principled splitting methods in segmentation tasks.

## 2    Related Work

Existing methodologies for stratifying complex datasets trace their origins to Iterative Stratification (IS) by [23]. In their work, the authors addressed the intricacies of multi-label datasets, proposing a greedy algorithm that iteratively assigns samples to folds based on the required frequency of positive labels within each fold. Fold quality is evaluated by comparing the Label Distribution (LD) metric. LD is the difference between the proportion of samples belonging to a given class within a fold and the corresponding proportion in the overall dataset

The partitioning method based on stratified random sampling (PMBSRS) by [24] employs a similar iterative approach but incorporates a pre-sorting step. This method clusters samples with similar label sets before assigning them equally to folds, thereby enhancing the homogeneity of label distribution across folds. Building upon the work of [23], authors of [25] introduced Second Order Iterative Stratification (SOIS), which considers second-order label relationships. Instead of focusing on individual labels, SOIS accounts for label pairs to determine the demand within each fold, thereby capturing more complex interactions between labels.

Other strategies approach stratification by starting from an initial assignment and iteratively refining it. The Stratified Sampling (SS) method proposed by [26] pre-assigns samples to folds and evaluates them based on deviations from the ideal label distribution. Samples contributing to the highest deviation subset are shuffled between folds to minimize discrepancies. This approach was designed with large-scale XML datasets in mind [31]. A similar split-and-shuffle approach was adopted by [27] but from a label-centric perspective. They developed a scoring mechanism to assess fold quality relative to each class, shuffling samples associated with the class exhibiting the poorest quality before recalculating scores. This method demonstrated effectiveness on large Gene Ontology datasets.

Finally, Evosplit, proposed by [28], leverages a genetic algorithm to address the challenges outlined by [23] and [25]. This method generates a population of fold assignments, evaluates their fitness, performs crossover and mutation, and iterates over generations to arrive at an optimal solution. Fitness is determined using Label Distribution (LD) from [23] and Label Pair Distribution (LPD) from [25].

While these methods offer valuable strategies for stratifying multi-label datasets, they do not directly extend to image segmentation. In segmentation, each sample contains a dense array of pixel-wise labels rather than a simple set of associated classes, and the proportion of pixels per class varies significantly across images. This makes stratification more complex: it is not enough to ensure the presence of a class in a fold; one must also consider how extensively each class is represented at the pixel level. Therefore, segmentation-specific stratification requires reformulating both the problem and the evaluation criteria. In the next section, we formalize the segmentation stratification task and introduce two new approaches, Iterative Pixel Stratification (IPS) and Wasserstein-Driven Evolutionary Stratification (WDES), which aim to create representative folds by explicitly accounting for pixel-level label distributions.

## 3    Stratification algorithms

Consider an image segmentation dataset $D = \{\mathbf{x}_n, \mathbf{y}_n\}_{n=1}^N$ with $N$ samples, where $\mathbf{x}_n$ denotes the input images and $\mathbf{y}_n$ the corresponding label masks. Let the total number of pixels across all samples be $P$, and the number of semantic classes be $C$. The objective is to partition $D$ into $K$ disjoint subsets, denoted $\mathbf{S} = (S^1, \ldots, S^K)$, according to a target proportion vector $\mathbf{r} = (r^1, \ldots, r^K)$, where $r^k$ specifies the desired proportion of samples in fold $k$.

Finding such a partition that preserves the global class distribution across folds is NP-Hard because it can be reduced to a multi-way partition problem [32]. It is analogous to dividing a multi-set of weighted elements into $K$ subsets with balanced sums; except here, the "weights" are pixel counts per class and the balance is across multiple classes simultaneously

Let $N_c$ be the number of samples in the dataset that contain class $c$, and $N_c^k$ the required number of such samples in fold $k$. Similarly, let $P^n$ be the total number of pixels in sample $n$, and $P_c^n$ the number of pixels of class $c$ in sample $n$. The total number of pixels of class $c$ in the dataset is $P_c$, and the desired number in fold $k$ is $P_c^k$. These quantities are central to the evaluation of fold quality and to guiding the stratification process.

## 3.1 Similarity measures

We first introduce similarity measures calculated on the subsets to assess the quality of the stratified folds **S**. These statistical properties of the subsets aim to quantify: (1) how well each fold adhere to the target sample proportions defined by **r**; (2) how closely the class-wise pixel proportions in each subset match those in the full dataset; (3) the overall similarity between fold-level and global label distributions using distributional distance.

**Sample Distribution (*SD*)**  With this measure, we aim to assess the deviation of the number of samples in each subset from the required number of samples as defined by the proportion vector **r**. For example, if a dataset of 10 samples is to be equally divided but the subsets contain 4 and 6 samples respectively, the value SD is 1. It is calculated as the average across folds of the deviations from the desired number of samples in each subset. A low SD indicates that the number of samples per fold closely matches the intended proportions.

$$\text{SD} = \frac{1}{K} \sum_{k=1}^{K} ||S^k| - N^k| \tag{1}$$

where $N^k = r^k \cdot N$ is the expected number of samples in fold $k$.

**Pixel Label Distribution (*PLD*)**  This measure evaluates how closely the proportion of pixels of each class within each subset matches the proportion of that class in the entire dataset. For each class $c$, it calculates the absolute difference between the ratio of pixels of that label in each subset $k$ and the ratio of pixels of that label in the whole dataset $D$. It then averages these results across all labels. This measure is inspired by Label Distribution (LD) in [23]. Lower PLD values indicate better preservation of class proportions at the pixel level.

$$\text{PLD} = \frac{1}{C} \sum_{c=1}^{C} \left( \frac{1}{K} \sum_{k=1}^{K} \left| \frac{P_c^k}{P_c - P_c^k} - \frac{P_c}{P - P_c} \right| \right) \tag{2}$$

An additional similarity measure examining class co-occurrence relationships is provided in Appendix H, where we extend this evaluation to capture second-order interactions between classes using the Pixel Label Pair Distribution (PLPD).

**Label Wasserstein Distance (*LWD*)**  This measure uses the Wasserstein Distance to quantify the dissimilarity between the cumulative class distributions in each fold and that of the entire dataset. By calculating this distance for each subset and averaging the results, LWD provides an indication of how closely the pixel-level class distributions in the subsets resemble that of the original dataset. Let $F_c$ and $F_c^k$ denote the cumulative pixel distributions up to class $c$ in the dataset and in fold $k$, respectively:

$$F_c = \sum_{i=1}^{c} P_i, \qquad F_c^k = \sum_{i=1}^{c} P_i^k$$

Then the Label Wasserstein Distance is defined as:

$$\text{LWD} = \frac{1}{K} \sum_{k=1}^{K} \sum_{c=1}^{C} \left| F_c - F_c^k \right| \tag{3}$$

To complement the Wasserstein-based similarity assessment, Appendix H presents further similarity assessments using alternative distributional measures, namely the Linear-kernel Maximum Mean Discrepancy (L-MMD) and the Linear Energy Distance (L-ED).

## 3.2 Iterative Pixel Stratification (*IPS*)

The motivation behind Iterative Pixel Stratification (IPS) stems from Iterative Stratification (IS) [23], which aims to distribute samples with specific positive labels proportionally across folds. This distribution is guided by two key properties within each fold: the number of samples required and the number of positive label instances required. To achieve this, IS starts by identifying the rarest label among the unassigned samples, determines which fold most requires positive instances of it and assigns samples that have positive instances of this label to that fold. In cases where multiple folds have equal demand, the fold with the fewest total samples is prioritized. Once a sample is assigned to a fold, it is removed from the pool of unassigned samples, and the demand for all positive labels associated with that sample is reduced in the fold. This iterative process continues, with the demand for specific positive labels decreasing as samples are assigned. The process concludes once all samples have been allocated, typically after C iterations.

IPS differs from IS in one key way: the label demand in folds is determined by the number of pixels required, not by the positive instances of that label. This also means that when a sample is assigned to a fold, the desire for all labels is reduced by the number of pixels of every label in that sample. As a result, IPS emphasizes achieving pixel-level balance over maintaining sample count proportionality. Note that IPS is implemented by adapting Iterative Stratification without additional optimizations thereby serving as a straightforward baseline. The pseudo-code for IPS algorithm is presented in Appendix B. An alternative iterative stratification approach is presented in Appendix G.

## 3.3 Wasserstein-Driven Evolutionary Stratification (*WDES*)

WDES formulates the stratification problem as an optimization task and solves it using a genetic algorithm. Each candidate solution, or *individual*, represents an assignment of samples to folds. Initially, a population of individuals is generated by randomly assigning samples to folds according to the target proportion vector **r**. Each *gene* (element) in an individual encodes the fold assignment of a specific sample.

The evolution of individuals is driven by crossover and mutation. First, we evaluate fitness using LWD (Eq. 3) and perform tournament selection [33]. Crossover occurs with a fixed probability, exchanging gene segments between individuals. A correction step follows to maintain target proportions across folds, reassigning samples if necessary. Mutation swaps assignments of randomly selected pairs, preserving fold proportions and adding diversity. This cycle repeats for a set number of generations, with the fittest individual (i.e., with the lowest LWD) selected as the final stratification.

We use the LWD as the fitness function to measure class distribution similarity. The proportionality requirement is managed through the initial population and crossover design, ensuring the genetic algorithm optimizes class distribution while maintaining proportionality across folds for a balanced stratification.

## 3.4 On the convergence of stratification algorithms

For large multi-label datasets, as the dataset size grows, random splits asymptotically preserve the marginal label frequencies in each fold due to the law of large numbers. We provided a mathematical proof of this property in Appendix A.1. This property makes random stratification a practical choice for very large datasets, where computational efficiency and simplicity are preferable.

For smaller datasets, where $N$ is finite, WDES provides strong guarantees of approaching the optimal stratification as the number of generations and population size increase.

**Proposition 3.1** (Convergence Rate of WDES to Empirical Optimum). *Let $\mathcal{F}_{M,G} \subset \mathcal{F}$ denote the set of stratifications explored by the WDES algorithm after $G$ generations with a population size $M$. Then the expected suboptimality of the final stratification $\mathcal{S}_{WDES} \in \mathcal{F}_{M,G}$ satisfies:*

$$\mathbb{E}\left[L(\mathcal{S}_{WDES})\right] - L^*_{\mathcal{F}_{M,G}} \leq \mathcal{O}\left(\frac{1}{MG}\right).$$

*where $L^*_{\mathcal{F}_{M,G}}$ denotes the optimal Wasserstein score in the empirical subset.*

*Proof.* See Appendix A.2. $\qquad\qquad\square$

The above results shows that the expected quality of the WDES stratification improves at a rate of O(1/MG), meaning that as the number of generations and population size increase, the WDES algorithm converges toward the global optimum. This makes WDES a particularly effective choice for small datasets, where the algorithm can afford the additional computational cost required to reach an optimal stratification, providing guarantees of high-quality splits as the search progresses.

# 4 Experiments

We conduct a comparative analysis of three stratification strategies: a) random sampling using KFold from the `scikit-learn` library [34], b) Iterative Pixel Stratification, and c) Wasserstein-Driven Evolutionary Stratification implemented using the `deap` library [35].

To achieve this, we start by testing these algorithms on the datasets outlined in Section 4.1 to calculate the statistical similarity measures (described in Section 3.1). For all the algorithms, we consider the number of pixels belonging to each class in the provided mask of every sample to guide the stratification procedure. The applicable parameters used for WDES are outlined in Appendix D. We perform stratification with datasets randomly shuffled five times and average the calculated similarity measures.

Following this, we perform 10-fold cross-validation tests with random splitting and the targeted stratification strategies. For every fold, we train a UNet [6] with a `resnet34` encoder from [36] for 50 epochs to perform segmentation on the images. The training process employs a learning rate of `2e-4` and utilizes the `Adam` optimizer with `DiceLoss`. For PascalVOC, `CELoss` is used instead, with a learning rate of `1e-4` and trained for 100 epochs. All experiments are conducted on a single Nvidia A100 graphics card with 40GB of VRAM, without distributed training, ensuring consistent and comparable results across the different stratification methods. We evaluate the stratification methods by measuring the variance in model performance across folds, to assess whether targeted stratification provides more stable and reliable evaluations.

## 4.1 Datasets

We select datasets spanning four major application domains: autonomous driving/street scenes, medical images, satellite imagery, and general-purpose datasets. For autonomous driving, we use Cityscapes [21] and CamVid [18]. In the medical domain, we consider EndoVis2018, a robotic scene segmentation dataset from MICCAI 2018 [19]. For satellite imagery, we use the LoveDA dataset [20], which focuses on land-cover segmentation in urban and rural locations. For general-purpose segmentation tasks, we include PascalVOC 2012 [17].

To characterize the complexity of each dataset, we define several key properties. Class Cardinality (CC) refers to the average number of classes present in a sample. Class Ubiquity (CU) captures the average number of samples in which each class appears, indicating how frequently each class occurs across the dataset. Average Imbalance Ratio (AIR) is the mean of class-wise imbalance ratios, where the imbalance ratio for a given class is the ratio of pixels in the most frequent class to those in the class under consideration. Entropy quantifies the diversity in class proportions; higher entropy implies a more balanced distribution of classes, whereas lower entropy indicates imbalance. These metrics provide a nuanced understanding of the datasets' structural diversity and segmentation challenges. The properties of all datasets are summarized in Table 1, sorted by entropy.

Table 1: Datasets and properties

| Dataset | N Images | N Classes | CC | CU | AIR | Entropy |
|---------|----------|-----------|------|-------|----------|---------|
| PascalVOC [17] | 2,913 | 21 | 2.48 | 344 | 79 | 1.82 |
| CamVid [18] | 701 | 32 | 17.44 | 370 | 7.56E+06 | 1.94 |
| EndoVis [19] | 2,235 | 12 | 6 | 1,117 | 1,492 | 2.38 |
| LoveDA [20] | 2,522 | 8 | 4.97 | 1568 | 4.5 | 2.61 |
| CityScapes [21] | 2,974 | 35 | 16.6 | 1,412 | 467 | 3.17 |

Table 2: Comparison of stratification methods using statistical similarity measures across datasets. WDES yields folds that better reflect the overall distribution, as indicated by PLD and LWD. The advantage narrows for datasets with higher class cardinality and lower ubiquity.

| Dataset | Method | SD Mean | SD Std | PLD $(\times 10^{-5})$ Mean | PLD $(\times 10^{-5})$ Std | LWD $(\times 10)$ Mean | LWD $(\times 10)$ Std |
|---------|--------|------|------|------|------|------|------|
| PascalVOC | Random | **0.42** | **0.00** | 955 | 137 | 75.0 | 2.62 |
| | IPS | 10.28 | 2.46 | 733 | 244 | 53.3 | 1.20 |
| | WDES | **0.42** | **0.00** | **456** | **77.9** | **51.4** | **1.63** |
| Camvid | Random | **0.00** | **0.00** | 63.4 | 3.73 | 183 | 5.08 |
| | IPS | 6.68 | 1.18 | 76.6 | 5.30 | 209 | 8.44 |
| | WDES | **0.00** | **0.00** | **36.7** | **4.47** | **138** | **7.84** |
| EndoVis | Random | **0.50** | **0.00** | 609 | 81.4 | 661 | 35.5 |
| | IPS | 11.42 | 1.05 | 769 | 12.2 | 764 | 75.2 |
| | WDES | **0.50** | **0.00** | **208** | **23.8** | **399** | **10.4** |
| LoveDA | Random | **0.32** | **0.00** | 1110 | 109 | 1080 | 88.4 |
| | IPS | 8.06 | 1.15 | 1090 | 207 | 1100 | 87.4 |
| | WDES | **0.32** | **0.00** | **377** | **31.4** | **635** | **16.0** |
| Cityscapes | Random | **0.50** | **0.00** | 126 | 15.4 | 304 | 16.5 |
| | IPS | 11.54 | 2.69 | 148 | 17.4 | 324 | 22.2 |
| | WDES | **0.50** | **0.00** | **67** | **3.44** | **217** | **5.17** |
| Ranking | Random | **1** | | 2.4 | | 2.2 | |
| | IPS | 2 | | 2.6 | | 2.8 | |
| | WDES | **1** | | **1** | | **1** | |

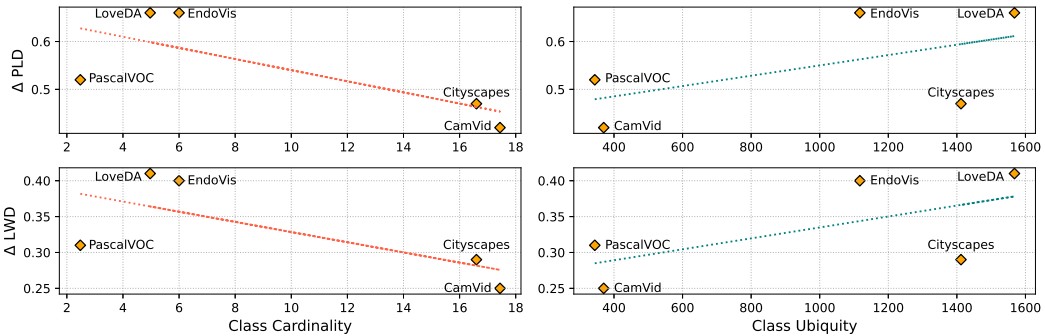

Figure 2: Relative improvement of WDES over random splitting in PLD and LWD. Performance gap decreases with increasing class cardinality, but WDES maintains an edge in datasets with high class ubiquity.

Table 3: Average segmentation performance of UNet across stratification methods in 10-fold cross-validation. Standard deviations in Accuracy, F1, and IoU are used to assess consistency. WDES performs best in low-entropy datasets, while random splitting becomes competitive as entropy increases.

| Dataset | Method | Accuracy | | F1 | | IoU | |
|---|---|---|---|---|---|---|---|
| | | Mean | Std $(\times 10^{-3})$ | Mean | Std $(\times 10^{-3})$ | Mean | Std $(\times 10^{-3})$ |
| PascalVOC | Random | 0.76 | 20.6 | 0.58 | 32.5 | 0.44 | 34.3 |
| | IPS | 0.76 | 14.5 | 0.58 | 30.3 | 0.44 | 31.6 |
| | WDES | 0.75 | **11.0** | 0.57 | **24.0** | 0.43 | **24.2** |
| Camvid | Random | 0.94 | 0.68 | 0.91 | 1.16 | 0.89 | 1.17 |
| | IPS | 0.94 | 0.74 | 0.91 | 1.35 | 0.89 | 1.27 |
| | WDES | 0.94 | **0.67** | 0.91 | **1.09** | 0.89 | **1.06** |
| EndoVis | Random | 0.94 | 19.4 | 0.86 | 27.0 | 0.80 | 28.1 |
| | IPS | 0.94 | 14.1 | 0.87 | 29.3 | 0.79 | 30.7 |
| | WDES | 0.94 | **13.0** | 0.85 | **17.9** | 0.79 | **18.9** |
| LoveDA | Random | 0.88 | **6.63** | 0.80 | **6.39** | 0.69 | **8.38** |
| | IPS | 0.88 | 7.39 | 0.80 | 9.29 | 0.69 | 10.7 |
| | WDES | 0.88 | 7.49 | 0.80 | 9.86 | 0.69 | 11.0 |
| Cityscapes | Random | 0.79 | 11.2 | 0.61 | **9.12** | 0.50 | **8.64** |
| | IPS | 0.79 | **10.3** | 0.61 | 14.9 | 0.50 | 15.2 |
| | WDES | 0.79 | 11.2 | 0.61 | 14.4 | 0.50 | 12.8 |
| Ranking | Random | 2.4 | | 1.8 | | 1.8 | |
| | IPS | 2 | | 2.6 | | 2.6 | |
| | WDES | **1.6** | | **1.6** | | **1.6** | |

## 5 Results and Discussion

### 5.1 Distribution of Labels and Samples

Table 2 presents the statistical similarity measures calculated for the subsets generated across the five datasets. For each measure and dataset, we report the mean over five runs, highlighting the best-performing method in bold. An overall average rank is also computed for each method by assigning rank 3 to the method with the lowest mean and rank 1 to the one with the highest.

Our findings indicate that WDES consistently achieves superior performance in terms of PLD and LWD across all datasets, underscoring its effectiveness in preserving label proportions and label distributions within the stratified subsets. Although IPS was explicitly designed to address proportionality in class presence, WDES outperforms it not only in LWD (which it directly optimizes) but also in PLD. This reinforces the perspective that stratification for image segmentation tasks benefits more from minimizing distributional dissimilarity than from simply ensuring class proportions.

WDES also exhibits favorable SD scores, which follow directly from its design. Because WDES explicitly enforces predefined sample proportions across folds, it naturally results in balanced sample counts. This behavior is similarly observed in random splitting, which allocates samples proportionally across folds without replacement.

Another noteworthy pattern emerges in relation to dataset complexity. As class cardinality increases, the performance gap between WDES and random splitting diminishes for both PLD and LWD. This trend suggests that when more classes are present per sample, the benefits of targeted stratification decrease due to the inherent balancing effect of high cardinality. In contrast, as class ubiquity increases, indicating that classes are more commonly present across the dataset, the superiority of WDES becomes more pronounced. This is expected, as higher ubiquity leads to more overlap between classes and samples, thereby amplifying the benefits of optimization-based stratification. These trends are illustrated in Figure 2, which presents the changes in PLD and LWD differences relative to class cardinality and ubiquity using trend lines.

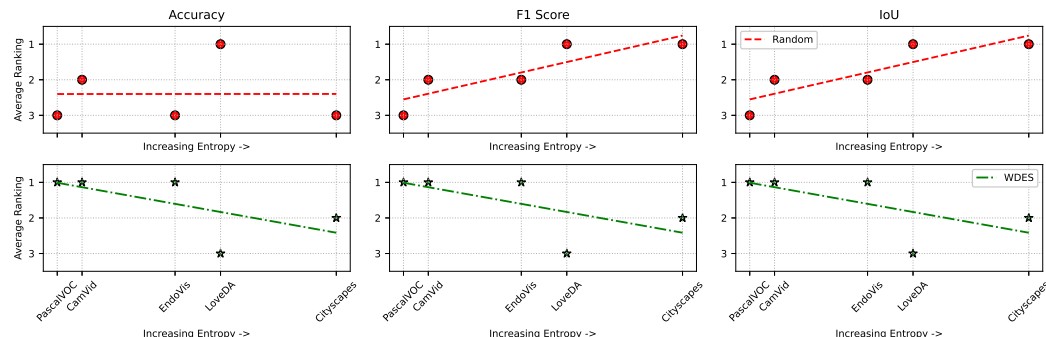

Figure 3: Stratification method rankings by performance variance across datasets. WDES ranks highest in low-entropy settings; random splitting improves with higher entropy, reducing the benefit of targeted stratification.

Table 4: Average segmentation performance of UNet for underrepresented classes (*bicycle*, *boat*, and *potted plant*) in PascalVOC under 10-fold cross-validation. Standard deviations in Accuracy, F1 Score, and IoU are reported to assess consistency. WDES yields the most stable results across all three rare classes, indicating improved fold balance for low-frequency categories.

| Class | Method | Accuracy | | F1 | | IoU | |
|---|---|---|---|---|---|---|---|
| | | Mean | Std | Mean | Std | Mean | Std |
| Bicycle | Random | 0.62 | 0.04 | 0.31 | 0.08 | 0.19 | 0.06 |
| | IPS | 0.62 | 0.04 | 0.31 | 0.09 | 0.19 | 0.06 |
| | WDES | 0.62 | **0.03** | 0.31 | **0.07** | 0.19 | **0.05** |
| Boat | Random | 0.72 | 0.05 | 0.53 | 0.09 | 0.37 | 0.08 |
| | IPS | 0.70 | 0.06 | 0.49 | 0.14 | 0.33 | 0.12 |
| | WDES | 0.72 | **0.04** | 0.55 | **0.08** | 0.38 | **0.08** |
| Potted Plant | Random | 0.71 | 0.08 | 0.47 | 0.16 | 0.32 | 0.14 |
| | IPS | 0.71 | 0.09 | 0.51 | 0.16 | 0.36 | 0.14 |
| | WDES | 0.70 | **0.07** | 0.47 | **0.12** | 0.31 | **0.10** |

## 5.2 Variance of Performance

Table 3 presents the results of training a UNet model on each dataset using 10-fold cross-validation. For each fold, we evaluate performance using Accuracy, F1 Score, and Intersection over Union (IoU), and report the average across the ten experiments. To highlight the robustness of each stratification strategy, we identify the method with the lowest standard deviation for each metric. Rankings are assigned from 1 to 3, with 1 indicating the lowest variance in performance and thus better fold consistency.

Our analysis shows that WDES achieves the best average rank in standard deviation across Accuracy, F1 Score, and IoU. This indicates that WDES produces splits that are more consistent with each other and better reflect the overall data distribution, leading to more stable model evaluation. While we cannot distinguish whether the reduced variance stems from the training or test set, the smaller size of the test set suggests it is more affected by sampling variability. Thus, the improvements in test set consistency are likely the main driver of the reduced standard deviation. This makes the average cross-validation score under WDES a more reliable estimate of a model's true generalization performance. We leave empirical validation of this to future work.

To further examine whether these improvements extend to underrepresented categories, we analyzed the segmentation performance for three rare classes in PascalVOC: *bicycle*, *boat*, and *potted plant*. As shown in Table 4, WDES consistently yields lower standard deviation in Accuracy, F1 Score, and IoU for these classes compared to random and IPS-based splits. This mirrors the global trend

observed in our main results, suggesting that WDES not only improves stability across folds overall but also enhances evaluation consistency for rare classes in low-entropy datasets.

As we move from low-entropy to high-entropy datasets, where label distributions are inherently more balanced and diverse, the advantage of WDES diminishes. The performance of random splitting improves and can even outperform stratified approaches. In these cases, the class distributions are already relatively uniform across samples, making targeted stratification less critical. This trend is reflected in the increasing rank of random splitting and the decline in WDES performance with increasing entropy, as illustrated in Figure 3.

While IPS was designed to be a label-aware stratification method, it ranks lowest on average across accuracy, F1, and IoU deviation. A potential reason lies in its single-pass, greedy allocation strategy, which reduces label demands in a fold as soon as a sample is assigned. This mechanism introduces a path dependency that prevents the algorithm from reassessing earlier decisions or accounting for broader distributional goals. Furthermore, although IPS considers pixel-level proportions, its strategy is primarily focused on satisfying per-label pixel quotas without evaluating the resulting inter-label distribution in each fold. As a result, by prioritizing rare labels first and assigning samples based solely on immediate label demand, it may overlook more nuanced strategies for dividing the dataset, such as balancing co-occurring label groups, preserving contextual diversity, or accounting for interdependencies between frequent and infrequent classes.

## 6    Conclusion and Limitations

In this paper, we introduced WDES, a stratification method for image segmentation tasks based on a genetic algorithm that minimizes the Wasserstein distance between label distributions. We provide a theoretical guarantee (Appendix A.2) that WDES converges to a globally optimal stratification given sufficient generations. We empirically compared its performance against iterative (IPS) and random sampling across several datasets and evaluation criteria. WDES outperforms both iterative and random methods, achieving the best average rank in accuracy, F1 score, and IoU variance in segmentation tasks, particularly for low-entropy datasets. Conversely, random stratification shows lower variance in high-entropy datasets, indicating that the benefits of targeted stratification diminish as class distributions become more uniform.

We emphasize the importance of stratification in cross-validation and train-test splits for image segmentation, particularly to avoid underrepresenting certain test sets and biasing the evaluation. This issue is especially critical for rare or underrepresented classes and when working with small datasets. Because final performance scores in cross-validation are typically computed as the mean of fold-wise means, the effects of imbalanced splits are not mitigated by the repeated nature of the process. While WDES improves upon simpler strategies, it is not a universal solution. Its reliance on minimizing Wasserstein distance assumes that distributional similarity alone ensures balanced splits, potentially overlooking other factors such as spatial structure or class co-occurrence. Future work should explore alternative similarity measures and more comprehensive modeling of dataset composition to improve stratification in segmentation tasks.

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

# A   Convergence of Stratification Algorithms

This section presents two results on the convergence behavior of stratification methods. The first shows that random splits of large multi-label datasets asymptotically preserve marginal label frequencies in each fold, by the law of large numbers, ensuring representativeness without explicit stratification. The second analyzes WDES, an evolutionary algorithm that minimizes label distribution differences across folds. It proves that the expected quality of the best-found stratification improves at a rate of O(1/MG), converging toward the global optimum as the number of generations and population size grow.

## A.1   Asymptotic Representativeness of Random Splits in Multi-Label Datasets

**Theorem A.1.** *Let $D = \{(\mathbf{x}_n, \mathbf{y}_n)\}_{n=1}^{N}$ be a dataset where each $\mathbf{y}_n \in \{0,1\}^L$ is a binary label vector over a fixed set of $L$ labels. Suppose $D$ is randomly partitioned into $K$ disjoint folds $\mathcal{S} = \{S^1, \ldots, S^K\}$ such that each fold satisfies $|S^k| \approx r^k N$, for a fixed target proportion vector $\mathbf{r} \in [0,1]^K$ with $\sum_{k=1}^{K} r^k = 1$. Then, for any label index $\ell \in \{1, \ldots, L\}$, the empirical label frequency in each fold converges almost surely to the true marginal label frequency as $N \to \infty$. That is, for all $k = 1, \ldots, K$,*

$$\hat{p}_N^k(\ell) \xrightarrow{a.s.} p(\ell),$$

*where*

$$\hat{p}_N^k(\ell) = \frac{1}{|S^k|} \sum_{n \in S^k} y_{n\ell} \quad and \quad p(\ell) = \mathbb{E}[y_{n\ell}] = \mathbb{P}(y_{n\ell} = 1).$$

*Proof.* Fix a label index $\ell \in \{1, \ldots, L\}$. For each sample $n$, define the binary indicator:

$$Z_n^{(\ell)} = y_{n\ell} \in \{0, 1\}.$$

By assumption, the data points $(\mathbf{x}_n, \mathbf{y}_n)$ are drawn i.i.d., so the sequence $\{Z_n^{(\ell)}\}_{n=1}^{N}$ is i.i.d. Bernoulli with mean:

$$\mathbb{E}[Z_n^{(\ell)}] = p(\ell).$$

Now consider the random assignment of samples to each fold $S^k$, where $|S^k| \approx r^k N$. For sufficiently large $N$, each $|S^k| \to \infty$ and the samples in each fold remain representative of the full dataset.

Let

$$\hat{p}_N^k(\ell) = \frac{1}{|S^k|} \sum_{n \in S^k} Z_n^{(\ell)}.$$

Then, by the Strong Law of Large Numbers (SLLN),

$$\hat{p}_N^k(\ell) \xrightarrow{a.s.} \mathbb{E}[Z_n^{(\ell)}] = p(\ell), \quad \text{as } |S^k| \to \infty.$$

Therefore, for any $k, k' \in \{1, \ldots, K\}$,

$$\left| \hat{p}_N^k(\ell) - \hat{p}_N^{k'}(\ell) \right| \xrightarrow{a.s.} 0,$$

and each fold becomes asymptotically representative of the true label distribution.    □

## A.2   Optimality and Convergence of WDES

To quantify the effectiveness of WDES, we derive a convergence bound that characterizes how the quality of stratifications improves with the number of generations and population size under standard evolutionary algorithm assumptions.

**Proposition A.1** (Convergence Rate of WDES to Empirical Optimum). *Let $\mathcal{F}_{M,G} \subset \mathcal{F}$ denote the set of stratifications explored by the WDES algorithm after $G$ generations with a population size $M$. Then the expected suboptimality of the final stratification $\mathcal{S}_{WDES} \in \mathcal{F}_{M,G}$ satisfies:*

$$\mathbb{E}\left[L(\mathcal{S}_{WDES})\right] - L^*_{\mathcal{F}_{M,G}} \leq \mathcal{O}\left(\frac{1}{MG}\right).$$

*where $L^*_{\mathcal{F}_{M,G}}$ denotes the optimal Wasserstein score found in the empirical subset.*

*Proof.* WDES is a genetic algorithm that evolves a population of $M$ stratifications over $G$ generations. The algorithm operates using elitist selection, where the best individual is preserved across generations. This ensures that the sequence of best-found scores $\{L_{\text{best}}^g\}_{g=1}^G$ is non-increasing.

Each generation performs $M$ fitness evaluations, totaling $MG$ evaluations across the run. The search space $\mathcal{F}$ is finite because there are finitely many possible assignments of $N$ labeled data points to $K$ folds satisfying approximate size constraints. Hence, WDES induces a finite-state stochastic process over $\mathcal{F}_{M,G} \subset \mathcal{F}$, the set of stratifications visited during execution.

Let $D = \{(\mathbf{x}_n, \mathbf{y}_n)\}_{n=1}^N$ be a dataset with $C$ classes, and let $\mathcal{F}$ denote the space of feasible stratifications $\mathcal{S} = \{S^1, \ldots, S^K\}$, where each fold $S^k$ satisfies the approximate size constraint $|S^k| \approx r^k N$, for a target proportion vector $\mathbf{r} \in [0,1]^K$, with $\sum_k r^k = 1$.

Define the class distribution in fold $k$ as $P_i^k = \frac{1}{|S^k|} \sum_{n \in S^k} \mathbb{I}[y_n = i]$, and the global class distribution as $P_i = \frac{1}{N} \sum_{n=1}^N \mathbb{I}[y_n = i]$, for $i = 1, \ldots, C$. Let $F_c^k = \sum_{i=1}^c P_i^k$ and $F_c = \sum_{i=1}^c P_i$ be the cumulative class distributions in fold $k$ and the full dataset, respectively.

The Label Wasserstein Distance of stratification $\mathcal{S}$ is defined as:

$$L(\mathcal{S}) = \frac{1}{K} \sum_{k=1}^K \sum_{c=1}^C \left| F_c^k - F_c \right|.$$

Crucially, the mutation operator used in WDES is assumed to be ergodic, i.e., it has a positive probability of reaching any feasible stratification from any other. This property, together with elitist selection, guarantees that the algorithm converges in probability to the global optimum over $\mathcal{F}$, as shown in [37]. However, in finite time, WDES only explores $\mathcal{F}_{M,G}$, and its best possible result is $L_{\mathcal{F}_{M,G}}^*$. The expected gap between the final stratification's score and this empirical optimum decreases with the number of independent evaluations $MG$. This follows from classical convergence results for evolutionary algorithms under elitist, ergodic dynamics, where the convergence rate to the best solution seen is:

$$\mathbb{E}\left[L(\mathcal{S}_{\text{WDES}})\right] - L_{\mathcal{F}_{M,G}}^* \leq \mathcal{O}\left(\frac{1}{MG}\right).$$

This bound reflects the diminishing returns of increasing population size and generations, but confirms that more search budget leads to better empirical solutions. □

**Remark.** *The theorem guarantees convergence to the best stratification within the explored subset $\mathcal{F}_{M,G}$, not necessarily the global optimum $L^* = \min_{\mathcal{S} \in \mathcal{F}} L(\mathcal{S})$. Nonetheless, due to the ergodic nature of the mutation operator, the probability of reaching any region of the space is non-zero. Thus, $L_{\mathcal{F}_{M,G}}^* \to L^*$ as $G \to \infty$, and the algorithm converges to the global optimum in probability.*

# B   Iterative Pixel Stratification Algorithm

Following the notation introduced in Section 3, the algorithm for IPS is described in Algorithm 1.

---

**Algorithm 1** Iterative Pixel Stratification

---

**Input**: A dataset $D$ of $C$ classes, $N$ samples that have a total of $P$ pixels. The dataset is to be divided into $K$ folds in the proportion $\mathbf{r} = r_1, \ldots, r_k$.

**Output**: Disjoint subsets $S_1, \ldots, S_k$

1:  # Calculate desired number of samples per fold
2:  $\{N^k\}_1^K \leftarrow N \cdot \mathbf{r}$
3:  **for** $c \leftarrow 1$ **to** $C$ **do**
4:      # Calculate number of pixels present per class
5:      $P_c \leftarrow \{P : c \in C\}$
6:      # Calculate desired number of pixels per class per fold
7:      $\{P_c^k\}_1^K \leftarrow P_c \cdot \mathbf{r}$
8:  **end for**
9:  **while** $|D| > 0$ **do**
10:     # Find rarest class
11:     $l \leftarrow \underset{c}{\operatorname{argmin}}\, P_c$
12:     **for** $(\mathbf{x}_l, \mathbf{y}_l) \in D_l$ **do**
13:         # Find folds with largest desire for class $l$
14:         $M \leftarrow \underset{k \in K}{\operatorname{argmax}}\, P_l^k$
15:         **if** $|M| = 1$ **then**
16:             $m \leftarrow M$
17:         **else**
18:             $M' \leftarrow \underset{k \in M}{\operatorname{argmin}}\, N^k$
19:             **if** $|M'| = 1$ **then**
20:                 $m \leftarrow M'$
21:             **else**
22:                 $m \leftarrow$ Random Element of $(M')$
23:             **end if**
24:         **end if**
25:         # Assign sample to subset and remove from dataset
26:         $S^m \leftarrow S^m \cup (\mathbf{x}_l, \mathbf{y}_l)$
27:         $D \leftarrow D \setminus \{(\mathbf{x}_l, \mathbf{y}_l)\}$
28:         $N^m \leftarrow N^m - 1$
29:         **for** $c \in C$ **do**
30:             $P_c^m = P_c^m - P_c(\mathbf{x}_l, \mathbf{y}_l)$
31:         **end for**
32:     **end for**
33: **end while**

---

# C  Wasserstein-Driven Evolutionary Stratification Algorithm

Following the notation introduced in Section 3, the algorithm for WDES is described in Algorithm 2.

---

**Algorithm 2** Wasserstein-Driven Evolutionary Stratification (WDES)

---

     **Input**: A dataset $D$ of $C$ classes and $N$ samples, to be divided into $K$ folds according to proportions $\mathbf{r} = (r_1, \ldots, r_K)$; population size $M$; number of generations $G$; crossover rate $p_c$; mutation rate $p_m$.

     **Output**: Disjoint subsets $S_1, \ldots, S_K$ corresponding to the fittest individual.

  1: `# Initialize population of candidate stratifications`
  2: **for** $i \leftarrow 1$ **to** $M$ **do**
  3:     Generate random fold assignments according to $\mathbf{r}$
  4:     Store as individual $I_i$
  5: **end for**
  6: **for** $g \leftarrow 1$ **to** $G$ **do**
  7:     `# Evaluate fitness of each individual`
  8:     **for** $i \leftarrow 1$ **to** $M$ **do**
  9:         Evaluate fitness using Label Wasserstein Distance (Eq. 3) for fold assignments in $I_i$
10:     **end for**
11:     `# Select parents via tournament selection`
12:     Select pairs of individuals based on lowest LWD values
13:     `# Apply crossover and correction`
14:     **for** $p \leftarrow 1$ **to** number of parent pairs **do**
15:         **if** random() $< p_c$ **then**
16:             Exchange random segments of fold assignments between parents
17:             Apply correction to preserve fold proportions $\mathbf{r}$
18:         **end if**
19:     **end for**
20:     `# Apply mutation`
21:     **for** $i \leftarrow 1$ **to** $M$ **do**
22:         **if** random() $< p_m$ **then**
23:             Randomly swap fold assignments of two samples in $I_i$
24:         **end if**
25:     **end for**
26: **end for**
27: `# Select best-performing individual as final stratification`
28: $I^* \leftarrow \underset{I_i}{\arg\min} \, \mathrm{LWD}(I_i)$
29: Return $S_1, \ldots, S_K$ as the folds encoded in $I^*$

---

## D    Genetic Algorithm parameters

We empirically selected the hyper-parameters for the genetic algorithm by evaluating performance across multiple datasets. For each parameter, we increased its value until no further improvements were observed in the stratification quality. The final values are shown in Table 5,

Table 5: WDES Parameters

| Parameter | Value |
| --- | --- |
| Number of Generations | 50 |
| Number of Individuals | 100 |
| Gene Mating Probability | 0.5 |
| Individual Mutation Probability | 0.2 |
| Selection Tournament size | 3 |

## E    Time and Space Complexity of WDES

The computational complexity of the Wasserstein Distance-based Evolutionary Stratifier (WDES) can be analyzed in terms of the population size $M$, number of generations $G$, number of samples $N$, number of classes $C$, and number of folds $K$. Each individual in the population represents an assignment of all $N$ samples into $K$ folds, and the fitness evaluation involves computing the Wasserstein distance across $C$-dimensional class histograms.

The overall time complexity is therefore $\mathcal{O}(M \cdot G \cdot K \cdot C)$, with an additional linear component in $N$ arising from crossover and mutation operations. The space complexity is $\mathcal{O}(M \cdot N)$, as the algorithm maintains a population of $M$ individuals, each encoding fold assignments for all samples.

In practice, the runtime of WDES is further influenced by the genetic algorithm parameters, including mutation and crossover probabilities, which determine how much of the population is updated between generations. These hyperparameters affect convergence speed and computational cost but do not alter the asymptotic complexity. An empirical assessment of runtime performance is provided in the following section.

## F    Runtime Analysis

We evaluate the runtime performance of our stratification methods on an Apple MacBook Pro with an M3 Pro processor and 36 GB of memory. Figure 4 presents the runtime in seconds (with a logarithmic scale on the y-axis) for Random splitting, IPS, and WDES across all datasets.

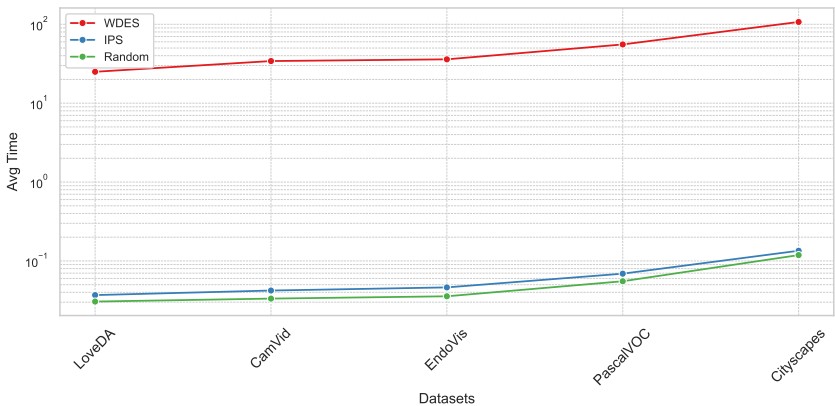

Figure 4: Runtime comparison of Random splitting, IPS, and WDES across all datasets.

Table 6: Comparison of stratification methods on similarity-based evaluation metrics. Results are averaged across datasets for Random splitting, IPS, CS-IPS, and WDES. While CS-IPS performs slightly better than IPS in some cases, WDES consistently achieves the best scores across all metrics, indicating stronger distributional alignment.

| Dataset | Method | SD | | PLD $(\times 10^{-5})$ | | LWD $(\times 10)$ | |
|---|---|---|---|---|---|---|---|
| | | Mean | Std | Mean | Std | Mean | Std |
| PascalVOC | Random | **0.42** | **0.00** | 955 | 137 | 75.0 | 2.62 |
| | CS-IPS | 0.96 | 0.00 | 525 | 6.33 | 42.5 | 0.32 |
| | IPS | 10.28 | 2.46 | 733 | 244 | 53.3 | 1.20 |
| | WDES | **0.42** | **0.00** | **456** | **77.9** | **51.4** | **1.63** |
| Camvid | Random | **0.00** | **0.00** | 63.4 | 3.73 | 183 | 5.08 |
| | CS-IPS | 0.60 | 0.00 | 58.0 | 0.39 | 175 | 1.48 |
| | IPS | 6.68 | 1.18 | 76.6 | 5.30 | 209 | 8.44 |
| | WDES | **0.00** | **0.00** | **36.7** | **4.47** | **138** | **7.84** |
| EndoVis | Random | **0.50** | **0.00** | 609 | 81.4 | 661 | 35.5 |
| | CS-IPS | 0.80 | 0.00 | 485 | 8.00 | 549 | 3.34 |
| | IPS | 11.42 | 1.05 | 769 | 12.2 | 764 | 75.2 |
| | WDES | **0.50** | **0.00** | **208** | **23.8** | **399** | **10.4** |
| LoveDA | Random | **0.32** | **0.00** | 1110 | 109 | 1080 | 88.4 |
| | CS-IPS | 0.64 | 0.00 | 935 | 11.9 | 892 | 7.58 |
| | IPS | 8.06 | 1.15 | 1090 | 207 | 1100 | 87.4 |
| | WDES | **0.32** | **0.00** | **377** | **31.4** | **635** | **16.0** |
| Cityscapes | Random | **0.50** | **0.00** | 126 | 15.4 | 304 | 16.5 |
| | CS-IPS | 1.90 | 0.00 | 130 | 4.48 | 300 | 4.73 |
| | IPS | 11.54 | 2.69 | 148 | 17.4 | 324 | 22.2 |
| | WDES | **0.50** | **0.00** | **67** | **3.44** | **217** | **5.17** |
| Ranking | Random | **1** | | 3.0 | | 3.2 | |
| | CS-IPS | 2 | | 2.2 | | 2.0 | |
| | IPS | 3 | | 3.8 | | 3.8 | |
| | WDES | **1** | | **1** | | **1** | |

## G  Class-Sensitive Iterative Pixel Stratification (*CS-IPS*)

The Class-Sensitive Iterative Pixel Stratification (CS-IPS) algorithm introduces an alternative iterative approach for constructing balanced folds based on pixel-level class distributions. The method begins by ranking all classes in the dataset according to their rarity, determined by the total number of pixels belonging to each class. Starting from the rarest class, all samples containing pixels of that class are identified and sorted in descending order based on the number of pixels associated with the current class. The sorted samples are then distributed across folds in a zig-zag manner to ensure a balanced representation of the class across partitions. This procedure is repeated iteratively for each subsequent class until all samples have been assigned. The resulting folds aim to maintain pixel-level balance while accounting for class frequency in a straightforward, iterative manner.

We report the results of CS-IPS alongside those obtained using random splitting, IPS, and WDES in Table 6. While CS-IPS achieves slightly improved performance compared to IPS in several cases, WDES consistently attains the highest overall performance across datasets. These results reinforce our motivation for adopting a more flexible, globally optimized approach such as the genetic algorithm employed in WDES.

The corresponding segmentation results, summarized in Table 7, exhibit a similar trend. CS-IPS provides measurable improvements over IPS, indicating the benefits of incorporating class sensitivity in the iterative process. However, WDES continues to outperform all other methods, underscoring the advantages of optimization-driven stratification over purely heuristic approaches.

Table 7: Average segmentation performance of UNet under 10-fold cross-validation using different stratification methods. Reported values correspond to Accuracy, F1 Score, and IoU. CS-IPS shows improved stability compared to IPS but remains below WDES, which continues to yield the most consistent and reliable fold performance.

| Dataset | Method | Accuracy | | F1 | | IoU | |
|---|---|---|---|---|---|---|---|
| | | Mean | Std $(\times 10^{-3})$ | Mean | Std $(\times 10^{-3})$ | Mean | Std $(\times 10^{-3})$ |
| PascalVOC | Random | 0.76 | 20.6 | 0.58 | 32.5 | 0.44 | 34.3 |
| | CS-IPS | 0.71 | 12.2 | 0.45 | 26.2 | 0.32 | 25.3 |
| | IPS | 0.76 | 14.5 | 0.58 | 30.3 | 0.44 | 31.6 |
| | WDES | 0.75 | **11.0** | 0.57 | **24.0** | 0.43 | **24.2** |
| Camvid | Random | 0.94 | 0.68 | 0.91 | 1.16 | 0.89 | 1.17 |
| | CS-IPS | 0.94 | 0.70 | 0.91 | 1.32 | 0.89 | 1.24 |
| | IPS | 0.94 | 0.74 | 0.91 | 1.35 | 0.89 | 1.27 |
| | WDES | 0.94 | **0.67** | 0.91 | **1.09** | 0.89 | **1.06** |
| EndoVis | Random | 0.94 | 19.4 | 0.86 | 27.0 | 0.80 | 28.1 |
| | CS-IPS | 0.92 | 13.6 | 0.86 | 21.2 | 0.78 | 21.4 |
| | IPS | 0.94 | 14.1 | 0.87 | 29.3 | 0.79 | 30.7 |
| | WDES | 0.94 | **13.0** | 0.85 | **17.9** | 0.79 | **18.9** |
| LoveDA | Random | 0.88 | **6.63** | 0.80 | **6.39** | 0.69 | **8.38** |
| | CS-IPS | 0.88 | 7.30 | 0.80 | 8.19 | 0.69 | 10.3 |
| | IPS | 0.88 | 7.39 | 0.80 | 9.29 | 0.69 | 10.7 |
| | WDES | 0.88 | 7.49 | 0.80 | 9.86 | 0.69 | 11.0 |
| Cityscapes | Random | 0.79 | 11.2 | 0.61 | **9.12** | 0.50 | **8.64** |
| | CS-IPS | 0.79 | 12.2 | 0.61 | 21.5 | 0.51 | 18.2 |
| | IPS | 0.79 | **10.3** | 0.61 | 14.9 | 0.50 | 15.2 |
| | WDES | 0.79 | 11.2 | 0.61 | 14.4 | 0.50 | 12.8 |
| Ranking | Random | 2.6 | | 2.2 | | 2.2 | |
| | CS-IPS | 2.4 | | 2.6 | | 2.6 | |
| | IPS | 2.6 | | 3.2 | | 3.6 | |
| | WDES | **1.8** | | **1.8** | | **1.8** | |

# H   Additional Evaluation Metrics

To ensure that the evaluation of stratification quality is not biased toward a specific similarity metric, we further assess all stratification methods using additional distributional measures: the Linear-kernel Maximum Mean Discrepancy (L-MMD) and the Linear Energy Distance (L-ED). These metrics provide complementary perspectives on distributional similarity between folds and the overall dataset, capturing higher-order differences beyond those measured by the Label Wasserstein Distance (LWD).

Table 8 reports results for all methods—random splitting, IPS, CS-IPS, and WDES—across these alternative metrics. Consistent with the findings based on LWD, WDES achieves the lowest average discrepancy across datasets, demonstrating that its advantages generalize across diverse statistical criteria. IPS performs competitively on the PascalVOC dataset, which we attribute to its higher proportion of single-class samples. Overall, these results confirm that the performance gains of WDES are robust to the choice of similarity metric.

**Pixel Label Pair Distribution (*PLPD*)**   In addition to the aforementioned distributional measures, we incorporate a class co-occurrence metric inspired by the Label Pair Distribution (LPD) proposed in literature [25]. This measure captures second-order relationships between classes that tend to appear together within the same sample. We adapt this concept to the pixel level, computing the Pixel Label Pair Distribution (PLPD) to quantify the co-occurrence consistency between folds and the global dataset. The results, presented in Table 8, extend our analysis by evaluating how well each method preserves inter-class relationships. The previously introduced CS-IPS method is also included for completeness.

Table 8: Evaluation of stratification quality using alternative similarity measures: Pixel Label Pair Distribution (PLPD), Linear-kernel Maximum Mean Discrepancy (L-MMD), and Energy Distance (L-ED). Results are averaged across datasets for Random splitting, IPS, CS-IPS, and WDES. WDES consistently outperforms the other methods across all three metrics, demonstrating that its advantages generalize beyond Wasserstein-based evaluation.

| Dataset | Method | PLPD | L-MMD $(\times 10^2)$ | L-ED |
|---|---|---|---|---|
| PascalVOC | Random | 0.67 | 5.67 | 3.64 |
| | CS-IPS | 0.69 | 2.68 | 2.91 |
| | IPS | **0.66** | **1.84** | 2.56 |
| | WDES | 0.76 | 2.94 | **2.49** |
| Camvid | Random | 4.70 | 12.2 | 8.93 |
| | CS-IPS | 5.97 | 11.1 | 8.28 |
| | IPS | 4.62 | 14.7 | 9.91 |
| | WDES | **4.58** | **7.16** | **6.98** |
| EndoVis | Random | 0.36 | 46.8 | 14.7 |
| | CS-IPS | 0.29 | 35.7 | 11.7 |
| | IPS | 0.27 | 55.2 | 16.4 |
| | WDES | **0.25** | **17.9** | **9.53** |
| LoveDA | Random | $1.35_{\times 10^{-5}}$ | 78.6 | 20.9 |
| | CS-IPS | $1.22_{\times 10^{-5}}$ | 62.7 | 17.2 |
| | IPS | $1.21_{\times 10^{-5}}$ | 69.9 | 21.3 |
| | WDES | $\mathbf{1.20}_{\times 10^{-5}}$ | **27.7** | **12.8** |
| Cityscapes | Random | 3.73 | 20.7 | 7.70 |
| | CS-IPS | 3.75 | 21.2 | 7.44 |
| | IPS | 3.85 | 23.3 | 8.24 |
| | WDES | **3.68** | **11.8** | **5.79** |
| Ranking | Random | 3 | 3.2 | 3.2 |
| | CS-IPS | 3.2 | 2.2 | 2.2 |
| | IPS | 2.2 | 3.2 | 3.6 |
| | WDES | **1.6** | **1.4** | **1.0** |

These additional evaluations further corroborate the overall trends observed in the main results: while iterative approaches such as IPS and CS-IPS produce reasonable fold-level balance, WDES consistently achieves superior alignment with global label distributions across all tested metrics.

# I  Dataset details

## I.1  Cityscapes

Cityscapes is a large benchmark dataset for training and testing pixel-level and instance-level semantic labeling. It contains diverse stereo video sequences from street scenes in 50 cities. Of these frames, 5,000 images have high-quality pixel annotations across 30 visual classes, grouped into eight categories: flat, construction, nature, vehicle, sky, object, human, and void. We use only the left-camera images (the annotated view) and run our experiments on the training set, since it's the largest split.

## I.2  CamVid

The Cambridge-driving Labeled Video Database (CamVid) captures footage from the perspective of a driving automobile, unlike most videos that are filmed with fixed-position CCTV-style cameras. From this footage, 701 frames were sampled at 1 Hz and manually labeled with 32 semantic classes. Similar to CityScapes, we only use the largest subset, the training set.

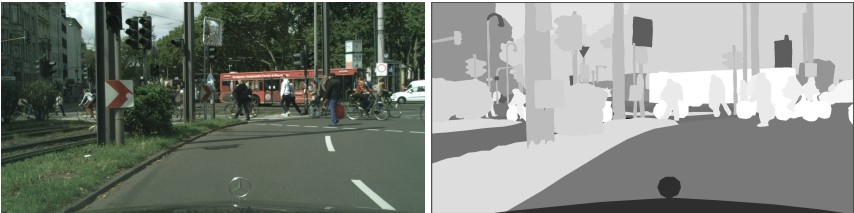

Figure 5: Example Cityscapes data: the original image (left) and its enhanced annotation mask (right), for better visualization.

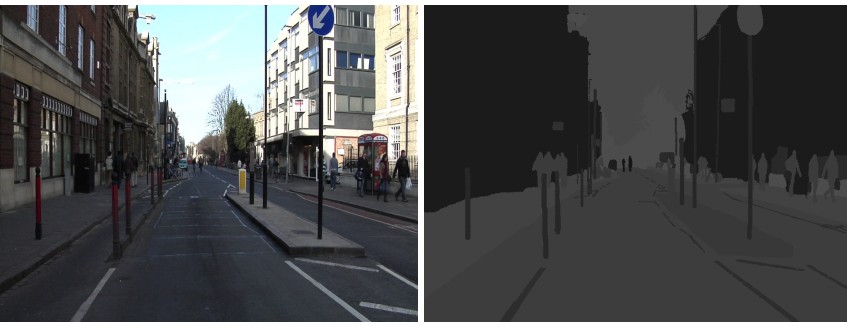

Figure 6: Example CamVid data: the original image (left) and its enhanced annotation mask (right).

### I.3 Pascal VOC 2012

The Pascal VOC 2012 serves as a cornerstone resource for training and comparing semantic segmentation models. It comprises 20 classes, including entities like people, animals, vehicles, and indoor objects. The dataset comprises 1,464 training images, 1,449 validation images, and a private testing set. Each image in this dataset is annotated with pixel-level segmentation, bounding box, and object class information. For our experiments, we exclusively utilized the training set.

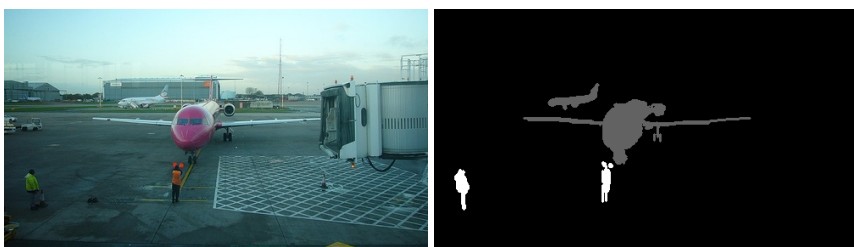

Figure 7: Example Pascal VOC 2012 data: the original image (left) and its enhanced annotation mask (right).

### I.4 EndoVis2018

The EndoVis2018 dataset, used in the Robotic Scene Segmentation Challenge of MICCAI 2018, contains 16 robotic nephrectomy procedures recorded using da Vinci Xi systems in porcine labs. Each procedure comprises 149 training frames and 250 testing frames, each with a resolution of 1280×1024. The dataset includes images from both the left and right eye cameras, as well as the stereo camera calibration parameters. However, labels are only available for the left eye camera, with 12 categories. In this case, we utilized only the training set.

### I.5 LoveDA

The LoveDA dataset, a comprehensive collection of remote sensing images, is designed for semantic segmentation. It comprises 5,987 high-resolution images captured across three Chinese cities,

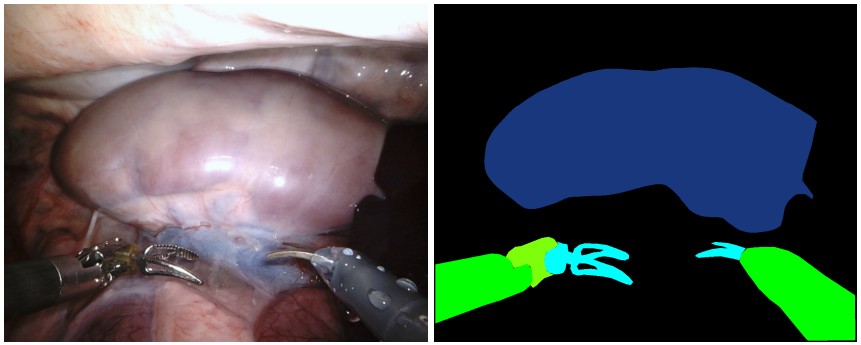

Figure 8: Example EndoVis2018 data: the original image (left) and its annotation mask (right).

capturing a diverse range of urban and rural scenes. These images are meticulously annotated with 166,768 object annotations, categorized into seven distinct land-cover classes: background, buildings, roads, water bodies, barren land, forests, and agricultural areas. Again, only the training set is utilized.

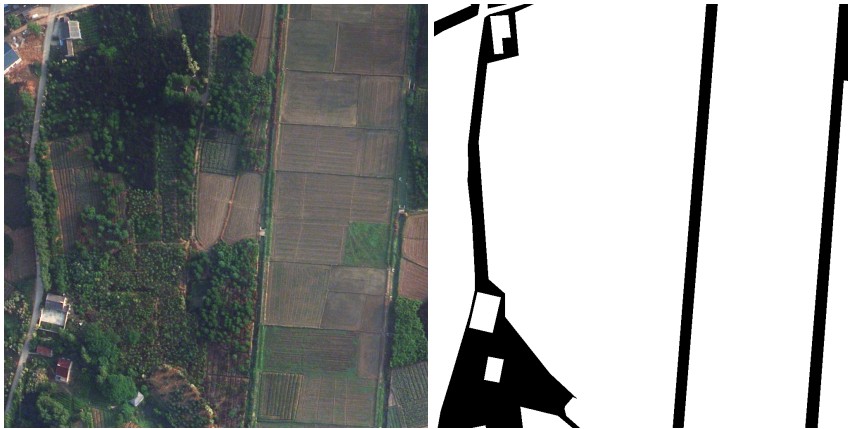

Figure 9: Example LoveDA data: the original image (left) and its enhanced annotation mask (right).

## J   Acknowledgments

The authors gratefully acknowledge the scientific support and HPC resources provided by the Erlangen National High Performance Computing Center (NHR) of the FAU Erlangen-Nürnberg. The hardware is funded by the Deutsche Forschungsgemeinschaft (DFG).

