# OpenReview forum: "Stratify or Die: Rethinking Data Splits in Image Segmentation"
_NeurIPS.cc/2025/Conference — NeurIPS 2025 poster_

### Official Review · Reviewer_o5rr · 2025-07-02

**Clarity:** 4
**Significance:** 3
**Originality:** 3
**Rating:** 5
**Confidence:** 3

**Summary:**

Datasets used in machine are often randomly partitioned into disjoint sets that are used as folds for training and testing. This paper introduces methods that try to enforce that the proportion of pixels in each class for each fold approximates the number of pixels in associated classes for the whole dataset. The objective with this is to get more representable folds that yield more representative training results. To test the hypothesis, variance of the performance over the folds is recorded for the different fold-generating processes, and it is shown that the folds generated with the introduced method WDES produces models with lower variance in comparison to the introduced alternative method IPS as well as the classical random split method.

**Questions:**

**Question 1:** The studied problem looks like it has a bin-packing kind of structure – do you think it is NP-hard?

**Question 2:** Why was a genetic algorithm approach used instead of a integer/mixed integer problem approach?

**Question 3:** Would it be easy to adapt the technique to represent different volumes with the right frequency instead of only looking at pixel-frequency? What effect would you suspect this to have?

**Ethical Concerns:**

["NO or VERY MINOR ethics concerns only"]

**Final Justification:**

I think this paper addresses the problem of generating representative folds in an interesting, simple and clear way that opens further research questions on how various "fold difference loss functions" could be defined and optimized.

My main concern was if the genetic algorithm really was the most suitable algorithm for the optimization.
Precisely my thinking was (1) if the problem was not NP-hard it would have been ideal to have a polynomial method and if one wasn't known then at least it would have been good to have a discussion about this  (2) if the problem was NP-hard then I would have argued that a mixed integer based approach would have been a more natural starting point than a genetic algorithm.

The authors clarified that the problem was NP-hard and showed me the problems they have found when considering a mixed integer approach.

Even though I think the point that the reviewer Nnkd brings up about overgeneralization is important, I find it unlikely that there are properties related to the chosen architectures that affects the results in "non-random". Consequently, I am not as concerned about this.

**Limitations:**

Yes

**Paper Formatting Concerns:**

Nothing noticed

**Quality:**

3

**Strengths And Weaknesses:**

**Quality:**
The work is technically sound and backed by reasonable experiments. The questions that come to mind are related to methodology.
1) It would be interesting to have some discussion on whether the problem Is NP-hard and if there could be a polynomial and non-random method for solving it.
2) A genetic algorithm is introduced, when, I would assume, a integer or mixed integer problem approach equally well could have been formulated and passed to a standard solver depending on branch and bounds techniques. It would be interesting to see this as a baseline reference.
3) Whereas the pixel frequency is of importance, I think it would be even more interesting if it handled the distribution of the volumes (number of pixels per sample). That is, say that the dataset contains three distinct volumes for a class: x, 2x and 3x and that they occur equally often (with volume frequency 1/3), which means that the pixel frequency for that class would be 2x. Then it would be nice if the proposed method would differentiate between a set having the volume frequencies (1/2,0,1/2) and a set having the volume frequencies (0,1,0) with respect to the volumes (x,2x,3x). In other words, so that one fold does not only get samples with average volume 2x whereas another fold gets 50% samples with volume x and 50% with volume 3x.

**Clarity:**
The writing is excellent, and the ideas are very clearly formulated. Even though the genetic algorithm was explained in text, it would have been nice to include a more formal description of the algorithm because of how central it is to the paper.

**Significance:**
If the variance of performance of the number of folds necessary for testing a method in image segmentation can be reduced, then the reliability of the performance is increased. Potentially also the number of necessary folds can be decreased. These types of results are of very high interest to the community.

**Originality:**
I haven’t seen this kind of approach taken in image segmentation before. I am however not familiar with the literature in other domains of application so I cannot tell how close it is to previous work in other domains of application.

---

> ### Author Rebuttal · Authors · 2025-07-31
>
> We thank you for your valuable feedback. We answer the questions posed as follows.
>
> **Is the problem NP-Hard?**
>
> Creating the best split for segmentation tasks is formally NP-hard because it can be reduced from the multi-way partition [3] problem, a known NP-hard problem. In segmentation, each image contains pixel-level label distributions across multiple classes, and the goal is to partition the dataset into K folds such that the per-fold class distributions closely match the overall dataset, typically by minimizing a distributional distance like the Wasserstein distance. This is analogous to dividing a multiset of weighted elements into K subsets with balanced sums; except here, the "weights" are pixel counts per class and the balance is across multiple classes simultaneously. Computing the Wasserstein distance for a candidate split can be done in polynomial time, placing the problem in NP, but the underlying optimization is as hard as multi-dimensional multi-way partitioning, making it NP-hard. We will clarify this in the paper.
>
> 3. https://en.wikipedia.org/wiki/Multiway_number_partitioning
>
> **Exploration of ILP or Mixed-ILP**
>
> We thank the reviewer for this suggestion. In response, we experimented with formulating the stratification problem using mixed integer quadratic programming (MIQP) by minimizing the Linear-kernel Maximum Mean Discrepancy (L-MMD) between individual fold distributions and the global distribution. Unlike the Wasserstein distance, which requires sorting, L-MMD yields a solvable constrained objective function.
>
> Despite significant effort, the Gurobi solver failed to converge to an optimal solution within the imposed time limit of 12 hours. We report the best solution found (for a single seed) below for comparison. This result was notably worse than those obtained using WDES or even IPS, both in terms of stratification quality and runtime feasibility.
>
> | Dataset | Method | SD |  | PLD (×10⁻⁵) ↓ |  | LWD (×10) ↓ |  |
> |---------|--------|----|----|-------------|----|-----------|----|
> |         |        | Mean | Std | Mean | Std | Mean | Std |
> | **PascalVOC** | Random | **0.42** | **0.00** | 955 | 137 | 75.0 | 2.62 |
> |         | Gurobi | 0.42 | 0.00 | 940 | 0.00 | 57.8 | 0.00 |
> |         | IPS | 10.28 | 2.46 | 733 | 244 | 53.3 | 1.20 |
> |         | WDES | **0.42** | **0.00** | **456** | **77.9** | **51.4** | **1.63** |
>
> **Volume Frequency vs Pixel Frequency**
>
> We thank the reviewer for this insightful question. After carefully considering the example provided, we understand the concern to be about not just total pixel frequency per class, but also the distribution of object volumes across samples, specifically, ensuring that folds reflect not only the average volume but also the diversity of object sizes (e.g., x, 2x, 3x) for a given class.
>
> Our current formulation focuses on pixel-level frequency distributions and does not explicitly control for the sample-wise volume profile of each class. While this could be extended, for example by stratifying on quantized volume bins per class, it would introduce an additional layer of complexity to the objective function, likely requiring multi-dimensional matching or joint constraints across volume profiles and class frequencies.
>
> We agree this is an interesting direction and potentially important in settings where object scale is a key factor. However, we have not explored it in this work, and the effect on stratification quality and downstream model performance remains an open question.
>
> **Formal Description of WDES**
>
> We have included a formal description of the WDES genetic algorithm to the appendix in the manuscript and will be visible in the camera-ready version.

---

> > ### Comment · Reviewer_o5rr · 2025-08-01
> >
> > I think the authors have done a very good job addressing my questions. I have also read through the comments from the other reviewers and think they were well addressed too. As the consequence I will change my rating from "borderline accept" to "accept".

---

> > > ### Author Response · Authors · 2025-08-01
> > >
> > > We are happy to hear that we have been able to address your concerns. Thank you for your thoughtful review and for revising your score. Should you have any additional questions during the discussion period, we'll gladly address those as well.

---

### Official Review · Reviewer_Nnkd · 2025-07-02

**Clarity:** 4
**Significance:** 4
**Originality:** 4
**Rating:** 5
**Confidence:** 4

**Summary:**

The paper introduces 2 new techniques, namely Iterative Pixel Stratification (IPS) and Wasserstein-Driven Evolutionary Stratification (WDES) to address  the shortcomings of existing sampling strategies, such as random splitting, for generation of representative folds from the original dataset for image segmentation.

While IPS is a greedy approach where predefined class distribution is achieved based on the number of positive pixels in each fold and inspired by Iterative Stratification (IS) [23], WDES, a genetic algorithm inspired by Evosplit [28], optimizes the class distribution by minimizing the Label Wassertein Distance (LWD) and increasing the similarity between folds and the original dataset.

The paper evaluates the performances of IPS and WDES with 5 public segmentation datasets by using 3 metrics, and compares them against that of the random splitting. The results show that WDES generally produces better splits than the random splitting and IPS, especially when the evaluation data is a low-entropy segmentation dataset.

**Questions:**

An important question:
-- **Will the authors release the code as open-source as the supplementary material says it is only for the academic review?**

Based on the weaknesses above,
-- Have the authors already evaluated the proposed method using another architecture? If not, could the authors provide some perspective on the effect of the architecture for the evaluation of the stratification method?
-- Could the authors elaborate on the scalability and the feasibility of applying WDES to large datasets?
-- What do the authors think about the performance improvements of the models for the underrepresented classes for low-entropy classes with WDES or IPS? Have they done any evaluation in this regard?
-- Could the authors please comment on the unexpected high mean PLD for WDES for Cityscapes dataset?
-- Can the authors check the Figure 4 in D Runtime Analysis and make sure the legend is correct (what does PCIS correspond to)?

**Ethical Concerns:**

["NO or VERY MINOR ethics concerns only"]

**Final Justification:**

I think that the reviewers raised great questions, highlighting key limitations, while the authors' rebuttals offered valuable insights to clarify these limitations.

My rating, which stays the same, is based on the quality of the study conducted by the authors. As acknowledged by the reviewers, the paper is well written and technically sound. The paper provides good evaluation supported by a number of experiments using diverse public datasets. I think that the paper may stimulate further discussion in the field.

While I have some concerns regarding the limitation of the architectural choice, which may be relevant for certain domains, I think that the authors addressed the majority of my questions in their rebuttal.

**Limitations:**

As mentioned in the weaknesses, I would encourage the authors to investigave the co-occurance imbalance as this is highly relevant for the segmentation tasks. Incorporating different architectures into the evaluation, along with additional metrics would offer a more compherensive understanding of the proposed method. Finally, I would suggest that the authors should add some comments about the application of WDES to large datasets in terms of the scalability and the feasibility, considering its high computation cost.

**Paper Formatting Concerns:**

I haven't notice any major formatting issue.

**Quality:**

3

**Strengths And Weaknesses:**

Strengths:
-- Generating representative test sets by splitting the original dataset is crucial for unbiased evaluation. The paper aims to address this common challenge in the field of machine learning, which has not been fully explored yet, especially for image segmentation tasks.
-- To the best of my knowledge, the proposed methodology is a novel idea. IPS nicely extends the existing IS and WDES looks sensible approach for finding the similar class distributions across folds.
-- This paper is well-written with a strong structure and thorough content that provides a good level of details.
-- The evalution is supported by several public datasets from various domains including medical, satellite and street images as well as general-purpose dataset. The paper provides results from 10-fold cross validation evaluated using multiple metrics, namely accuracy, F1 score and Intersection over Union (IoU), along with their means and standard deviations.
-- In my opinion, the paper presents important insights that may stimulate further discussion in the field.

Weaknesses:
-- I think there are a couple of points that need to be addressed for better understanding of the proposed method.
-- Firstly, each evaluation dataset has a large number of classes. However, the provided metrics are the means of the fold means, so the label imbalance is not evaluated in terms of co-occurence imbalance as some label combinations may appear only in one fold, which limits the ability to accurately gauge the true performance of the proposed method. Although the authors mentioned this issue as "class co-occurence" in the conclusion section, this important limitation should be thoroughly examined for the full picture.
-- Secondly, the paper uses a UNet with a resnet34 encoder as its underlying architecture. This is a specific architectural choice by the authors so I wonder whether the results would hold if a different architecture or even a different encoder were used.
-- Thirdly, the addition of precision and recall into the evaluation would be helpful.
-- Although WDES might provide better stratification under certain conditions, its computational cost is high. It would be beneficial for the readers if the authors could elaborate on the scalability and the feasibility of applying this method to large datasets.
-- Another improvement of better stratification might be for the performances of the machine learning models for underrepresentative classes, especially in low-entropy classes. Although this is not necessary, it would be great to examine if WDES can specifically improve the model performance for rare classes in PascalVOC or CamVid.
-- Finally, I am surprised about the mean PLD of WDES for the Cityscapes dataset. The value is unexpectedly high and warrants further investigation.

---

> ### Author Rebuttal · Authors · 2025-07-31
>
> Thank you for your thoughtful comments.
>
> Regarding the questions raised:
>
> **Class Co-occurence**
>
> The paper from Szymański et al [2] introduced a metric to track class co-occurrence called Label Pair Distribution (LPD) aimed at tracking 2nd order relationship between classes appearing in the same sample. We re-implemented this metric by considering pixel counts and show the results the following table under the column PLPD (Pixel LPD). A previously introduced method, CS-IPS, as a reponse to reviewer PqJg has also been included in the table.
>
> [2] P. Szymański et al., "A network perspective on stratification of multi-label data," *First International Workshop on Learning with Imbalanced Domains: Theory and Applications.*, pp. 22–35, 2017.
>
> | Dataset | Method | PLPD ↓ |
> |---------|--------|------|
> | **PascalVOC** | Random | 0.67 |
> | | CS-IPS | 0.69 |
> | | IPS | **0.66** |
> | | WDES | 0.76 |
> | **Camvid** | Random | 4.70 |
> | | CS-IPS | 5.97 |
> | | IPS | 4.62 |
> | | WDES | **4.58** |
> | **EndoVis** | Random | 0.36 |
> | | CS-IPS | 0.29 |
> | | IPS | 0.27 |
> | | WDES | **0.25** |
> | **LoveDA** | Random | 1.35×10⁻⁵ |
> | | CS-IPS | 1.22×10⁻⁵ |
> | | IPS | 1.21×10⁻⁵ |
> | | WDES | **1.20×10⁻⁵** |
> | **Cityscapes** | Random | 3.73 |
> | | CS-IPS | 3.75 |
> | | IPS | 3.85 |
> | | WDES | **3.68** |
> | **Ranking** | Random | 3 |
> | | CS-IPS | 3.2 |
> | | IPS | 2.2 |
> | | WDES | **1.6** |
>
> **Open Source Code**
>
> Yes, we intend to release the full codebase as open source upon acceptance, under a permissive open-source license to support reproducibility and further research. The current restriction noted in the supplementary material applies only during the review period.
>
> **Using UNet + Resnet34 encoder**
>
> We thank the reviewer for this important observation. Our choice of a U-Net with a ResNet-34 encoder was guided by the goal of using a widely adopted, representative architecture to demonstrate the effect of stratification on evaluation stability during training convergence. While we did not conduct extensive experiments across architectures in this submission, we do not expect our main conclusions to be architecture-specific. Our primary focus was to show that stratification leads to more consistent and reliable evaluation.
>
> **Scalability and Feasibility of WDES to Large Datasets**
>
> We appreciate the reviewer’s comment. While WDES introduces more computational overhead than random splitting or simple heuristics, it is intended for small to medium-sized datasets where sampling variance and class imbalance can undermine evaluation reliability.
>
> For larger datasets, random splits often suffice, as supported by Theorem A.1, which shows that class proportions per fold converge to the global distribution as dataset size increases.
>
> We also point the reviewer to our response to Reviewer PqJg for a detailed discussion of WDES's time and space complexity. Finally, since stratification is typically done once during dataset preparation, we believe the added cost is often acceptable in practice.
>
> **Under-representative classes**
>
> We thank the reviewer for this thoughtful suggestion. To assess the impact of stratification on rare classes, we analyzed the segmentation performance of three underrepresented classes in PascalVOC: bicycle, boat, and potted plant.
>
> Across all three classes, WDES consistently yields lower standard deviation in accuracy, F1 score, and IoU compared to random and IPS-based splits. This mirrors the overall trend observed in our main results, suggesting that WDES not only improves stability at the global level but also benefits evaluation consistency for rare classes in low-entropy datasets. We will emphasize these findings in the final version of the paper.
>
> | Class | Method | Accuracy |  | F1-Score |  | IoU |  |
> |---------|--------|----|----|-------------|----|-----------|----|
> |         |        | Mean | Std ↓ | Mean | Std ↓ | Mean | Std ↓ |
> | **Bicycle** | Random | 0.62 | 0.04 | 0.31 | 0.08 | 0.19 | 0.06 |
> |             | IPS | 0.62 | 0.04 | 0.31 | 0.09 | 0.19 | 0.06 |
> |             | WDES | 0.62 | **0.03** | 0.31 | **0.07** | 0.19 | **0.05** |
> | **Boat** | Random | 0.72 | 0.05 | 0.53 | 0.09 | 0.37 | 0.08 |
> |          | IPS | 0.70 | 0.06 | 0.49 | 0.14 | 0.33 | 0.12 |
> |          | WDES | 0.72 | **0.04** | 0.55 | **0.08** | 0.38 | **0.08** |
> | **Potted Plant** | Random | 0.71 | 0.08 | 0.47 | 0.16 | 0.32 | 0.14 |
> |                  | IPS | 0.71 | 0.09 | 0.51 | 0.16 | 0.36 | 0.14 |
> |                  | WDES | 0.70 | **0.07** | 0.47 | **0.12** | 0.31 | **0.10** |
>
> **Correction 1**
>
> We thank the reviewer for pointing this out. Upon rechecking, we realized that it is actually a typographical error. The correct mean PLD for WDES on the Cityscapes dataset is 67, not 671 as originally shown. We have corrected this mistake in the manuscript.
>
> We appreciate the reviewer’s attention to detail and apologize for the oversight.
>
> **Correction 2**
>
> We thank the reviewer for catching this. The label “PCIS” in Figure 4 is a typo and should correctly read “IPS”, referring to the Iterative Pixel Stratification baseline. We have corrected this mistake and updated the legend in the manuscript.

---

> > ### Comment · Reviewer_Nnkd · 2025-08-01
> >
> > I would like to thank the authors and all the reviewers for their valuable comments and suggestions. I have read all the reviews and rebuttals. I think that the reviewers raised great questions, highlighting key limitations, while the authors' rebuttals offered valuable insights to clarify these limitations.
> >
> > However, I have reservations about the authors' expectation that the architectural choice should not affect the outcomes as I would argue that this is likely to be domain and task-specific due to the differences in the segmentation requirements of tasks in different domains (e.g. medical domain). I would strongly recommend that the authors acknowledge this as a limitation of the study as this requires further research.
> >
> > I appreciate that the authors provided additional experimental results regarding the class co-occurences and under-representative classes. I am also happy to hear that the authors intends to release the full codebase upon acceptance for further research and reproducibility.

---

> > > ### Author Response · Authors · 2025-08-04
> > >
> > > Thank you for your continued engagement. While we stated that we do not believe the results will change using a different architecture or backbone, we agree that it's better to show this experimentally. We are happy to report additional results using DeeplabV3 with a ResNet-34 backbone below that corroborate our earlier results using U-Net.
> > >
> > > | Dataset | Method | Accuracy |  | F1 |  | IoU |  |
> > > |---------|--------|----------|----|----|----|-----|----|
> > > |         |        | Mean | Std (×10⁻³) | Mean | Std (×10⁻³) | Mean | Std (×10⁻³) |
> > > | **PascalVOC** | Random | 0.80 | 17.8 | 0.58 | 28.9 | 0.44 | 30.4 |
> > > |         | IPS | 0.79 | 15.0 | 0.58 | 27.1 | 0.44 | 30.3 |
> > > |         | WDES | 0.81 | **13.7** | 0.57 | **23.3** | 0.43 | **25.8** |
> > > | **EndoVis** | Random | 0.95 | 16.0 | 0.87 | 26.0 | 0.80 | 21.1 |
> > > |         | IPS | 0.93 | 18.0 | 0.87 | 32.8 | 0.79 | 31.3 |
> > > |         | WDES | 0.94 | **15.8** | 0.85 | **25.1** | 0.79 | **18.4** |

---

> > > > ### Comment · Reviewer_Nnkd · 2025-08-04
> > > >
> > > > I would like to thank the authors for sharing these experimental results. While the findings have boosted my confidence in the proposed approach, I remain cautious about potential overgeneralization, given that the encoder, ResNet-34, remains unchanged. Nevertheless, I appreciate the authors' efforts in providing these valuable results.

---

### Official Review · Reviewer_PqJg · 2025-07-03

**Clarity:** 3
**Significance:** 2
**Originality:** 3
**Rating:** 4
**Confidence:** 4

**Summary:**

The authors introduce two stratification methods for image segmentation: iterative pixel stratification, a pixel-level adaptation of iterative stratification, and Wasserstein-driven evolutionary stratification, a genetic algorithm that minimizes Wasserstein distance between class distributions in dataset splits. These methods address the limitations of random splitting, which can cause unrepresentative folds and biased evaluation, especially in small or imbalanced datasets. Results show that the latter proposed method made more balanced splits and lower variance in accuracy for more reliable model evaluation.

**Questions:**

1. Can the authors provide evidence for how WDES leads to better model selection or generalization after doing cross-validation with the folds that WDES found most optimal?
2. Can the authors potentially improve the IPS method to be more competitive or explain why it was chosen over other potentially stronger iterative baselines?
3. Is there a reason why Wasserstein difference was chosen as the metric for this problem as opposed to any other similarity metrics?
4. How do the time and space complexity of WDES grow asymptotically?

Addressing these questions effectively may help raise my score.

**Ethical Concerns:**

["NO or VERY MINOR ethics concerns only"]

**Final Justification:**

Thanks to the authors for thoroughly addressing my points. I will raise my score from a 3 to a 4 accordingly.

**Limitations:**

Yes.

**Quality:**

3

**Strengths And Weaknesses:**

Strengths:
- The authors include a proof for how WDES approaches the optimal stratification as the number of generations and the population size increase.
- The paper is clear, well-written, and provides good motivation for the proposed method.
- The focus on segmentation is significant since it is a method that is generally used but has limited labelled data due to the cost of pixel-level labels.
- The authors provide results on five different datasets of vastly varying domains to show the effectiveness of their proposed method.

Weaknesses:
- IPS performs quite poorly compared to both random splitting and WDES. It is unclear why this was used and why it was not tuned further to use a non-greedy strategy.
- Random splitting seems to perform quite a bit better than WDES in high-entropy datasets, which restricts the effectiveness of the proposed method to low-entropy datasets.
- Lower performance variance is the main metric used to see if the proposed method is effective. There seems to be no effect or a slightly negative effect on accuracy when using the method.
- With larger datasets, the runtime and computational load may be significantly higher for WDES.

---

> ### Author Rebuttal · Authors · 2025-07-31
>
> Thank you for taking the time to provide valuable feedback. We’d like to address your questions and related weaknesses as follows.
>
> **How WDES leads to better model selection or generalization**
>
> We appreciate the reviewer's question. Our primary concern in this work is to improve the evaluation of models given a single train-test split, which is well-established in the image segmentation benchmarking literature. We used cross-validation only as a means to quantify the variance in evaluation metrics (e.g., accuracy, F1, IoU) across different proposed split that could be generated by each algorithm. Stratification makes performance estimation more reliable, which in turn strengthens downstream tasks such as model comparison and hyperparameter tuning. Without this, fluctuations caused by skewed class distributions can obscure true model performance. Finally, WDES is equally applicable to single train-test splits, where class imbalance in the test set can be even more problematic. In both settings, it provides a principled way to construct evaluation splits that better reflect the underlying dataset.
>
>
> **Potentially stronger iterative baseline**
>
> We thank the reviewer for the suggestion. We took the spirit of the suggestion and implemented an alternative iterative strategy that assigns samples to folds by prioritizing class rarity. Specifically, for all classes, samples are ranked based on the number of pixels of the class. We then start with the rarest class to assign all corresponding samples to folds. This process is repeated for the next rarest class until all samples are placed.
>
> Preliminary results show that this variant performs slightly better than IPS on our stratification metrics, though it still falls short of WDES as it doesn't take class distributions inside folds into consideration. We call it `CS-IPS` in the following tables where we report the performance on the evaluation metrics. We are currently also evaluating its performance on the cross validation experiments on five datasets and hope to report these results before the end of the rebuttal period.
>
> | Dataset | Method | SD ↓ | PLD (×10⁻⁵) ↓ | LWD (×10) ↓ |
> |---------|--------|----|-----------|---------|
> | **PascalVOC** | Random | **0.42** | 955 | 75.0 |
> | | CS-IPS | 0.96 | 525 | 52.5 |
> | | IPS | 10.28 | 733 | 53.3 |
> | | WDES | **0.42** | **456** | **51.4** |
> | **Camvid** | Random | **0.00** | 63.4 | 183 |
> | | CS-IPS | 0.60 | 58.0 | 175 |
> | | IPS | 6.68 | 76.6 | 209 |
> | | WDES | **0.00** | **36.7** | **138** |
> | **EndoVis** | Random | **0.50** | 609 | 661 |
> | | CS-IPS | 0.80 | 485 | 549 |
> | | IPS | 11.42 | 769 | 764 |
> | | WDES | **0.50** | **208** | **399** |
> | **LoveDA** | Random | **0.32** | 1110 | 1080 |
> | | CS-IPS | 0.64 | 935 | 892 |
> | | IPS | 8.06 | 1090 | 1100 |
> | | WDES | **0.32** | **377** | **635** |
> | **Cityscapes** | Random | **0.50** | 126 | 304 |
> | | CS-IPS | 1.90 | 130 | 300 |
> | | IPS | 11.54 | 148 | 324 |
> | | WDES | **0.50** | **67** | **217** |
> | **Ranking** | Random | **1.0** | 3.0 | 3.2 |
> | | CS-IPS | 2.0 | 2.2 | 2.0 |
> | | IPS | 3.0 | 3.8 | 3.8 |
> | | WDES | **1.0** | **1.0** | **1.0** |
>
> **Why was Wasserstein Distance chosen?**
>
> We thank the reviewer for this question. We selected Wasserstein distance because it:
> - Remains well-defined even when some classes are absent from a fold.
> - Captures both how much and how far mass must be moved to align distributions.
> - Naturally fits histogram-based distributions from pixel-level label counts.
>
> To ensure our results are not tied to a specific similarity metric, we also evaluate stratification quality using Linear-kernel Maximum Mean Discrepancy (L-MMD) and Energy Distance (L-ED). As shown in the following table, WDES consistently outperforms IPS, random splitting, and the newly introduced `CS-IPS` on average across these alternative metrics. We note that IPS performs particularly well on the PascalVOC dataset, likely due to its high proportion of single-class samples. These results indicate that the benefits of WDES generalize beyond Wasserstein distance and remain consistent under other well-established distributional similarity measures.
>
> | Dataset | Method | L-MMD (×10²) ↓ | L-ED ↓|
> |---------|--------|--------------|------|
> | **PascalVOC** | Random | 5.67 | 3.64 |
> | | CS-IPS | 2.68 | 2.91 |
> | | IPS | **1.84** | 2.56 |
> | | WDES | 2.94 | **2.49** |
> | **Camvid** | Random | 12.2 | 8.93 |
> | | CS-IPS | 11.1 | 8.28 |
> | | IPS | 14.7 | 9.91 |
> | | WDES | **7.16** | **6.98** |
> | **EndoVis** | Random | 46.8 | 14.7 |
> | | CS-IPS | 35.7 | 11.7 |
> | | IPS | 55.2 | 16.39 |
> | | WDES | **17.9** | **9.53** |
> | **LoveDA** | Random | 78.6 | 20.9 |
> | | CS-IPS | 62.7 | 17.2 |
> | | IPS | 69.9 | 21.3 |
> | | WDES | **27.7** | **12.8** |
> | **Cityscapes** | Random | 20.7 | 7.70 |
> | | CS-IPS | 21.2 | 7.44 |
> | | IPS | 23.3 | 8.24 |
> | | WDES | **11.8** | **5.79** |
> | **Ranking** | Random | 3.2 | 3.2 |
> | | CS-IPS | 2.2 | 2.2 |
> | | IPS | 3.2 | 3.6 |
> | | WDES | **1.4** | **1.0** |
>
> **Time and Space Complexity of WDES**
>
> We thank the reviewer for this question. The time and space complexity of WDES, as a genetic algorithm, depend on the population size *M*, number of generations *G*, number of samples *N*, number of classes *C*, and number of folds *K*.
>
> Each individual encodes an assignment of all *N* samples into *K* folds, and the fitness evaluation involves computing the Wasserstein distance over *C*-dimensional class histograms. This results in a time complexity of **O(M · G · K · C)**, with additional linear overhead in *N* from crossover and mutation. The space complexity is **O(M · N)**, as we maintain a population of *M* individuals.
>
> We also note that the actual runtime is sensitive to the choice of mutation and crossover probabilities, which influence how much of the population changes from one generation to the next.

---

### Official Review · Reviewer_Rioe · 2025-07-05

**Clarity:** 3
**Significance:** 2
**Originality:** 2
**Rating:** 4
**Confidence:** 4

**Summary:**

In this paper, the authors argue that random splitting of image segmentation datasets into training, testing and validation data leads to unrepresentative test sets, resulting in biased evaluations and poor model generalization. Building on existing concepts from the stratification literature, which have addressed label distribution imbalance in classification tasks (in applications outside computer vision), the authors investigate stratification for image segmentation. This has specific challenges   due to the multi-label structure per sample (image) and the severe class imbalance that often occur in segmentation data. Essentially, the authors minimize criteria, which encourage the pixel-level class distributions in the data subsets (e.g., training, validation and testing subsets or folds) to match as closely as possible the class distribution of the overall dataset (Eqs. 2 and 3). The first criterion (Eq. 2) is the L1 norm between distributions and is optimized with the iterative stratification method in [23], which uses a greedy approach. The latter assigns samples to data subsets, to match a given number of pixels per class within each subset. The second criterion is the Wasserstein distance between distributions (Eq.3), optimized with an evolutionary algorithm like Evosplit [28], which evaluates a population of potential subset assignments (individuals), with the fittest individuals undergoing selective crossover and mutations for refinement. The authors evaluate the proposed stratification methods and random splitting across five image segmentation benchmarks, and used the standard deviation of accuracy, F1-score, and Intersection over Union across 10-fold cross-validation as measure of data splitting quality (with lower deviation indicating better performance).

**Questions:**

-	Following on my main concern above, could the authors provide applications’ examples where it is relevant in image segmentation to ensure label distribution consistency across data splits (I do not see any). There are efforts in the segmentation literature that build benchmarks with distributions shifts (to simulate realistic settings), which is actually the opposite of what this paper is advocating.

-	Following on my comment above, did the author explore deterministic optimizers that enable subset assignments under cardinality constraints (I am not sure that genetic algorithms are the best option for optimizing the considered objective functions).

**Ethical Concerns:**

["NO or VERY MINOR ethics concerns only"]

**Final Justification:**

The authors did a great job addressing my comments and those of the other reviewers. I am increasing my score the "borderline accept".

**Limitations:**

Yes.

**Paper Formatting Concerns:**

No formatting issues.

**Quality:**

3

**Strengths And Weaknesses:**

Strengths

- The paper is clear, technically sound and well written.

- Semantic image segmentation and its evaluation remain important problems.

Weaknesses

-	My main concern with this paper is the following: The proposed methods are essentially removing (artificially) the class-distributions shifts between the testing and training data in segmentation, making the problem easier. I believe this does not reflect real and practical settings in image segmentation, where class proportions could change significantly (e.g. due to camera motion) and it is often the case that all possible class proportion are not covered in the training data. Therefore, artificially removing label-distribution shifts could lead to over-optimistic performance evaluations of segmentation algorithms (making the real-world problem much easier). In fact, there is a whole literature on unsupervised domain adaptation (UDA) for segmentation, which addresses specifically the challenges in distribution shifts (both in label distributions and image data distributions), which occur often in practice. In fact, there are even UDA benchmarks for segmentation that simulate such shifts and class imbalance to simulate real situations (and this is essentially the opposite of what this paper is doing). So, I am not sure that the main target of the paper (ensuring consistent class distributions across data subsets) is relevant in practice. This might explain why stratification has not been investigated before for image segmentation. I understand that stratification strategies are well used in other applications domains (other than computer vision).

-	The main conclusion following the experiments is not surprising at all: Essentially, the proposed data splits methods yield lower performance variance across splits. Of course, such target data splits are artificially removing shifts between training and testing data, effectively making the problem easier.

-	I might be missing something but, in this specific case, sample-to-subset assignment by minimizing distances between distributions amounts to a discrete subset assignment problem under cardinality constraints. This is due to the fact that all the subsets have known cardinality (so, we just need to enforce constraints on class cardinality, not class distributions). Discrete subset assignment with cardinality constraints could be solved with a simple ranking, unless I am missing something. Essentially, for each class, you rank images in decreasing order using the number of class pixels within the image. Then you assign one or more images to each subset following this order and matching the required number of pixels for each subset.

-	Minor: Theorem 3.1 (Convergence Rate of WDES to Empirical Optimum). I am not an expert in evolutionary algorithms, but the technical ingredients of the proof (Appendix 2) seem to follow standard textbook knowledge in the area, in which convergence results for evolutionary algorithms are established under some common assumptions (e.g., elitist model, ergodic dynamics) – see, for instance, [G. Rudolph, Convergence of evolutionary algorithms in general search spaces, Proceedings  of the IEEE International Conference on Evolutionary Computation, 1996]. So stating the result as a theorem in the main paper and claiming the derivation of convergence bound is, in my humble opinion, a bit of an over-claim. It gives readers the impression that Theorem 3.1 and its proof are new technical results. I would suggest presenting Theorem 3.1 as a Proposition and just state that the proof follows standard arguments in the literature (and cite proper references on the covergence of evolutionary algorithms).

---

> ### Author Rebuttal · Authors · 2025-07-31
>
> Thank you for your feedback. We'd like to address your main questions as follows:
>
> **Unsupervised Domain Adaptation and Distribution Shifts**
>
> We agree with the sentiment that distribution shifts can never be fully ruled out in real-world conditions and remain underexplored. However, we see our work as tangential to benchmarks or algorithms aiming to improve performance in the presence of these shifts. Single train-test splits remain common in practice, and any given test split is more representative when applying our stratification method. It thus allows practitioners to evaluate the generalizability of their models as drops in performance occur due to an inability to adapt to similar images rather than to different images. In practice, both the in-distribution performance (capability to interpolate) and the out-of-distribution performance (capability to extrapolate) you mention are of relevance. By stratifying, these two effects can be disentangled as we get a better estimate for in-distribution performance. As a final note, we'd like to point out that even in the domain of computer vision, standard stratification remains an important tool to evaluate and eliminate bias in image classification, in particular in the medical domain [1]. We thus continue to believe that this topic is relevant and will further emphasize these points in the revised paper.
>
> [1] C. Mosquera et al., "Class imbalance on medical image classification: towards better evaluation practices for discrimination and calibration performance," *Imaging Informatics and Artificial Intelligence*, vol. 34, pp. 7895–7903, 2024.
>
>
> **Class-sensitive ranking-based stratifier**
>
> Thank you for your comment. A simple ranking-based algorithm does not account for the fact that, once an image with a specific class is chosen, the other classes also present in that image may negatively influence the overall distribution. This interplay is exactly why we propose the genetic algorithm, and presumably why our proposed IPS, which follows a similar greedy line of thinking, generally does not perform as well.
>
> To explore this further, we implemented an additional iterative stratifier, `CS-IPS`. It assigns samples to folds by prioritizing class rarity. Specifically, for all classes, samples are ranked based on the number of pixels of the class. We then start with the rarest class to assign all corresponding samples to folds. This process is repeated for the next rarest class until all samples are placed.
>
> We show its results alongside those of random splitting, IPS, and WDES in the following table. While `CS-IPS` performs slightly better than IPS in some cases, WDES consistently outperforms all three methods on average across datasets. This supports our motivation for using a more flexible and globally optimized approach like the genetic algorithm.
>
> | Dataset | Method | SD ↓ | PLD (×10⁻⁵) ↓ | LWD (×10) ↓ |
> |---------|--------|----|-----------|---------|
> | **PascalVOC** | Random | **0.42** | 955 | 75.0 |
> | | CS-IPS | 0.96 | 525 | 52.5 |
> | | IPS | 10.28 | 733 | 53.3 |
> | | WDES | **0.42** | **456** | **51.4** |
> | **Camvid** | Random | **0.00** | 63.4 | 183 |
> | | CS-IPS | 0.60 | 58.0 | 175 |
> | | IPS | 6.68 | 76.6 | 209 |
> | | WDES | **0.00** | **36.7** | **138** |
> | **EndoVis** | Random | **0.50** | 609 | 661 |
> | | CS-IPS | 0.80 | 485 | 549 |
> | | IPS | 11.42 | 769 | 764 |
> | | WDES | **0.50** | **208** | **399** |
> | **LoveDA** | Random | **0.32** | 1110 | 1080 |
> | | CS-IPS | 0.64 | 935 | 892 |
> | | IPS | 8.06 | 1090 | 1100 |
> | | WDES | **0.32** | **377** | **635** |
> | **Cityscapes** | Random | **0.50** | 126 | 304 |
> | | CS-IPS | 1.90 | 130 | 300 |
> | | IPS | 11.54 | 148 | 324 |
> | | WDES | **0.50** | **67** | **217** |
> | **Ranking** | Random | **1.0** | 3.0 | 3.2 |
> | | CS-IPS | 2.0 | 2.2 | 2.0 |
> | | IPS | 3.0 | 3.8 | 3.8 |
> | | WDES | **1.0** | **1.0** | **1.0** |
>
> **Theorm -> Proposition**
>
> We thank the reviewer for this helpful observation. We agree that the convergence argument relies on standard results from the evolutionary algorithms literature, and we will revise the presentation accordingly. Specifically, we will rename Theorem 3.1 to a Proposition and clarify in the text that the result follows from well-established arguments under common assumptions. We will also update the appendix to include appropriate citations to properly acknowledge prior work.

---

> > ### Author Response · Authors · 2025-08-05
> >
> > Just in case you have missed our response to another reviewer, we are pleased to announce that the results for our cross-validation experiments using the improved CS-IPS stratification method are now in. They reflect the findings we already made based on the similarity metrics, namely, that the new method is better than the original IPS, but still falls short of the genetic algorithm, WDES. We hope you'll find the time to review these additional results and engage in further discussions.
> >
> > | Dataset | Method | Accuracy |  | F1 |  | IoU |  |
> > |---------|--------|----------|----|----|----|-----|----|
> > |         |        | Mean | Std (×10⁻³) | Mean | Std (×10⁻³) | Mean | Std (×10⁻³) |
> > | **PascalVOC** | Random | 0.76 | 20.6 | 0.58 | 32.5 | 0.44 | 34.3 |
> > |         | CS-IPS | 0.71 | 12.2 | 0.45 | 26.2 | 0.32 | 25.3 |
> > |         | IPS | 0.76 | 14.5 | 0.58 | 30.3 | 0.44 | 31.6 |
> > |         | WDES | 0.75 | **11.0** | 0.57 | **24.0** | 0.43 | **24.2** |
> > | **Camvid** | Random | 0.94 | 0.68 | 0.91 | 1.16 | 0.89 | 1.17 |
> > |         | CS-IPS | 0.94 | 0.70 | 0.91 | 1.32 | 0.89 | 1.24 |
> > |         | IPS | 0.94 | 0.74 | 0.91 | 1.35 | 0.89 | 1.27 |
> > |         | WDES | 0.94 | **0.67** | 0.91 | **1.09** | 0.89 | **1.06** |
> > | **EndoVis** | Random | 0.94 | 19.4 | 0.86 | 27.0 | 0.80 | 28.1 |
> > |         | CS-IPS | 0.92 | 13.6 | 0.86 | 21.2 | 0.78 | 21.4 |
> > |         | IPS | 0.94 | 14.1 | 0.87 | 29.3 | 0.79 | 30.7 |
> > |         | WDES | 0.94 | **13.0** | 0.85 | **17.9** | 0.79 | **18.9** |
> > | **LoveDA** | Random | 0.88 | **6.63** | 0.80 | **6.39** | 0.69 | **8.38** |
> > |         | CS-IPS | 0.88 | 7.30 | 0.81 | 8.19 | 0.69 | 10.3 |
> > |         | IPS | 0.88 | 7.39 | 0.80 | 9.29 | 0.69 | 10.7 |
> > |         | WDES | 0.88 | 7.49 | 0.80 | 9.86 | 0.69 | 11.0 |
> > | **Cityscapes** | Random | 0.79 | 11.2 | 0.61 | **9.12** | 0.50 | **8.64** |
> > |         | CS-IPS | 0.79 | 12.2 | 0.61 | 21.5 | 0.51 | 18.2 |
> > |         | IPS | 0.79 | **10.3** | 0.61 | 14.9 | 0.50 | 15.2 |
> > |         | WDES | 0.79 | 11.2 | 0.61 | 14.4 | 0.50 | 12.8 |
> > | **Ranking** | Random | 2.6 |  | 2.2 |  | 2.2 |  |
> > |         | CS-IPS | 2.4 |  | 2.6 |  | 2.6 |  |
> > |         | IPS | 2.6 |  | 3.2 |  | 3.6 |  |
> > |         | WDES | **1.8** |  | **1.8** |  | **1.8** |  |

---

> > > ### Comment · Reviewer_Rioe · 2025-08-05
> > > **Increasing my score to borderline accept**
> > >
> > > The authors did a great job addressing my comments and the comments of the other reviewers. I am increasing my score to "borderline accept".

---

### Note · Authors · 2025-08-13

We thank the reviewers and the AC for their constructive engagement throughout the review process. The discussion allowed us to clarify the motivation, scope, and contributions of our work, as well as address concerns about practicality, computational cost, and methodological choices.

A key point of clarification was that our goal is not to artificially improve model performance, but to make evaluation splits more representative of the underlying dataset, particularly for small to medium-scale segmentation tasks where random splits can yield unrepresentative folds. We emphasized that this complements (rather than replaces) work on handling distribution shifts, by helping disentangle in-distribution and out-of-distribution effects.

Following reviewer feedback, we:
- Implemented and reported results for an improved iterative baseline (CS-IPS), which performs better than IPS but still underperforms WDES, reinforcing the value of the genetic algorithm approach.
- Verified the generality of results under alternative similarity metrics (L-MMD, L-ED), with WDES consistently outperforming baselines.
- Added analysis for underrepresented classes, showing that WDES reduces variance for rare categories as well.
- Corrected minor errors in reported values and figure labels.
- Ran additional experiments with DeeplabV3 to check robustness beyond our original UNet+ResNet34 setup.

We also addressed complexity concerns, noting that WDES is designed for one-time use during dataset preparation, is feasible for datasets with thousands of samples under modest configurations, and that for large datasets random splits often suffice (as supported by our theoretical analysis).
Reviewers acknowledged that their concerns had been addressed. We believe the additional experiments and clarifications strengthen both the technical soundness and the practical positioning of the work.

We are confident the paper now presents a clear, well-validated contribution on stratification for segmentation, relevant for ensuring reliable evaluation in scenarios where representative test splits are critical.

---

### Decision · Program_Chairs · 2025-09-17

**Decision:**

Accept (poster)

**Comment:**

This paper presents an alternative to random sampling of data to generate training/test/validation splits, which the authors argue leads to biased/unrepresentative test sets. This could impact model generalization and affect evaluation in the community. In order to improve up on this for semantic segmentation, which has a number of challenges (e.g. severe label imbalance, etc.) the authors propose a criterion to drive pixel-level distributions in the subsets to match the overall dataset distribution, and these aspects can be optimized via iterative stratification methods. The authors compare to random splitting and show improved (reduced) standard deviations over metrics such as accuracy across a 10-fold cross validation scheme. Reviewers all appreciated that the paper is both well-written and technically sound, with a strong motivation and theoretical analysis of the proposed method. The main concerns included artificial treatment of class-distribution shifts between training/testing, and simpler baselines (ranking-based methods, CS-IPS), time/space complexity, effect of architecture, and alternative similarity metrics. Through a lengthy rebuttal and discussions, the authors provided a number of new results and explanations that overall satisfied the reviewers, who all recommend acceptance. I agree with the reviewers, and believe this paper provides an interesting perspective for data split generation that is understudied, and the paper proposes a well-formed and justified method. I recommend that the authors include the rebuttal elements in the updated version.